# INVARIANT CAUSAL REPRESENTATION LEARNING

## ABSTRACT

Due to spurious correlations, machine learning systems often fail to generalize to environments whose distributions differ from the ones used at training time. Prior work addressing this, either explicitly or implicitly, attempted to find a data representation that has an invariant causal relationship with the outcome. This is done by leveraging a diverse set of training environments to reduce the effect of spurious features, on top of which an invariant classifier is then built. However, these methods have generalization guarantees only when both data representation and classifiers come from a linear model class. As an alternative, we propose Invariant Causal Representation Learning (ICRL), a learning paradigm that enables out-of-distribution generalization in the nonlinear setting (i.e., nonlinear representations and nonlinear classifiers). It builds upon a practical and general assumption: data representations factorize when conditioning on the outcome and the environment. Based on this, we show identifiability up to a permutation and pointwise transformation. We also prove that all direct causes of the outcome can be fully discovered, which further enables us to obtain generalization guarantees in the nonlinear setting. Extensive experiments on both synthetic and real-world datasets show that our approach significantly outperforms a variety of baseline methods.

## 1 INTRODUCTION

In recent years, despite various impressive success stories, there is still a significant lack of robustness in machine learning algorithms. Specifically, machine learning systems often fail to generalize outside of a specific training distribution, because they usually learn easier-to-fit spurious correlations which are prone to change from training to testing environments. We illustrate this point by considering the widely used example of classifying images of camels and cows (Beery et al., 2018). The training dataset has a selection bias, i.e., most pictures of cows are taken in green pastures, while most pictures of camels happen to be in deserts. After training a convnet on this dataset, it is found that the model fell into the spurious correlation, i.e., it related green pastures with cows and deserts with camels, and therefore classified green pastures as cows and deserts as camels. The result is that the model failed to classify images of cows when they are taken on sandy beaches.

To address the aforementioned problem, a natural idea is to identify which features of the training data present domain-varying spurious correlations with labels and which features describe true correlations of interest that are stable across domains. In the example above, the former are the features describing the context (e.g., pastures and deserts), whilst the latter are the features describing animals (e.g., animal shape). Arjovsky et al. (2019) suggest that one can identify the stable features and build invariant predictors on them by exploiting the varying degrees of spurious correlation naturally present in training data collected from multiple environments. The authors proposed the invariant risk minimization (IRM) approach to find data representations for which the optimal classifier is invariant across all environments. Since this formulation is a challenging bi-leveled optimization problem, the authors proved the generalization of IRM across all environments by constraining both data representations and classifiers to be linear (Theorem 9 in Arjovsky et al. (2019)).

Ahuja et al. (2020) studied the problem from the perspective of game theory, with an approach that we call IRMG for short. They showed that the set of Nash equilibria for a proposed game are equivalent to the set of invariant predictors for any finite number of environments, even with nonlinear data representations and nonlinear classifiers. However, these theoretical results in the nonlinear setting only guarantee that one can learn invariant predictors from training environments, but do not guarantee that the learned invariant predictors can generalize well across all environments including

unseen testing environments. In fact, the authors directly borrowed the linear generalization result from Arjovsky et al. (2019) and presented it as Theorem 2 in Ahuja et al. (2020).

In this work we propose an alternative learning paradigm, called Invariant Causal Representation Learning (ICRL), which enables out-of-distribution (OOD) generalization in the nonlinear setting (i.e., nonlinear representations and nonlinear classifier). We first introduce a practical and general assumption: the data representation factorizes (i.e., its components are independent of each other) when conditioning on the outcome (e.g., labels) and the environment (represented as an index). This assumption builds a bridge between supervised learning and unsupervised learning, leading to a guarantee that the data representation can be identified up to a permutation and pointwise transformation. We then theoretically show that all the direct causes of the outcome can be fully discovered. Based on this, the challenging bi-leveled optimization problem in IRM and IRMG can be reduced to two simpler independent optimization problems, that is, learning the data representation and learning the optimal classifier can be performed separately. This further enables us to attain generalization guarantees in the nonlinear setting.

**Contributions**   We propose Invariant Causal Representation Learning (ICRL), a novel learning paradigm that enables OOD generalization in the nonlinear setting. (i) We introduce a conditional factorization assumption on data representation for the OOD generalization (Assumption 1). (ii) Base on this assumption, we show that each component of the representation can be identified up to a permutation and pointwise transformation (Theorem 1, 2 & 3). (iii) We further prove that all the direct causes of the outcome can be fully discovered (Proposition 1). (iv) We show that our approach has generalization guarantees in the nonlinear setting (Proposition 2). (v) Empirical results demonstrate that our approach significantly outperforms IRM and IRMG in the nonlinear scenarios.

## 2 PRELIMINARIES

### 2.1 IDENTIFIABLE VARIATIONAL AUTOENCODERS

A general issue with variational autoencoders[1] (VAEs) (Kingma & Welling, 2013; Rezende et al., 2014) is the lack of identifiability guarantees of the deep latent variable model. In other words, it is generally impossible to approximate the true joint distribution over observed and latent variables, including the true prior and posterior distributions over latent variables. Consider a simple latent variable model where $\boldsymbol{O} \in \mathbb{R}^d$ stands for an observed variable (random vector) and $\boldsymbol{X} \in \mathbb{R}^n$ for a latent variable. Khemakhem et al. (2020) showed that any model with unconditional latent distribution $p_{\boldsymbol{\theta}}(\boldsymbol{X})$ is unidentifiable. That is, we can always find transformations of $\boldsymbol{X}$ which change its value but do not change its distribution. Hence, the primary assumption that they make to obtain an identifiability result is to include a conditionally factorized prior distribution over the latent variables $p_{\boldsymbol{\theta}}(\boldsymbol{X}|\boldsymbol{U})$, where $\boldsymbol{U} \in \mathbb{R}^m$ is an additionally observed variable (Hyvarinen et al., 2019). More specifically, let $\boldsymbol{\theta} = (\boldsymbol{f}, \boldsymbol{T}, \boldsymbol{\lambda}) \in \Theta$ be the parameters of the conditional generative model:

$$p_{\boldsymbol{\theta}}(\boldsymbol{O}, \boldsymbol{X}|\boldsymbol{U}) = p_{\boldsymbol{f}}(\boldsymbol{O}|\boldsymbol{X})p_{\boldsymbol{T},\boldsymbol{\lambda}}(\boldsymbol{X}|\boldsymbol{U}), \tag{1}$$

where $p_{\boldsymbol{f}}(\boldsymbol{O}|\boldsymbol{X}) = p_{\boldsymbol{\epsilon}}(\boldsymbol{O} - \boldsymbol{f}(\boldsymbol{X}))$ in which $\boldsymbol{\epsilon}$ is an independent noise variable with probability density function $p_{\boldsymbol{\epsilon}}(\boldsymbol{\epsilon})$, and the prior probability density function is especifically given by

$$p_{\boldsymbol{T},\boldsymbol{\lambda}}(\boldsymbol{X}|\boldsymbol{U}) = \prod_i \mathcal{Q}_i(\boldsymbol{X}_i)/\mathcal{Z}_i(\boldsymbol{U}) \cdot \exp\big[\sum\nolimits_{j=1}^k T_{i,j}(\boldsymbol{X}_i)\lambda_{i,j}(\boldsymbol{U})\big], \tag{2}$$

where $\mathcal{Q}_i$ is the base measure, $\mathcal{Z}_i(\boldsymbol{U})$ the normalizing constant, $\boldsymbol{T}_i = (T_{i,1}, \ldots, T_{i,k})$ the sufficient statistics, $\boldsymbol{\lambda}_i(\boldsymbol{U}) = (\lambda_{i,1}(\boldsymbol{U}), \ldots, \lambda_{i,k}(\boldsymbol{U}))$ the corresponding parameters depending on $\boldsymbol{U}$, and $k$ the dimension of each sufficient statistic that is fixed in advance. It is worth noting that this assumption is not very restrictive as exponential families have universal approximation capabilities (Sriperumbudur et al., 2017). As in VAEs, we maximize the corresponding evidence lower bound:

$$\mathcal{L}_{\text{iVAE}}(\boldsymbol{\theta}, \boldsymbol{\phi}) := \mathbb{E}_{p_D}\big[\mathbb{E}_{q_{\boldsymbol{\phi}}(\boldsymbol{X}|\boldsymbol{O},\boldsymbol{U})}[\log p_{\boldsymbol{\theta}}(\boldsymbol{O}, \boldsymbol{X}|\boldsymbol{U}) - \log q_{\boldsymbol{\phi}}(\boldsymbol{X}|\boldsymbol{O},\boldsymbol{U})]\big], \tag{3}$$

where we denote by $p_D$ the empirical data distribution given by dataset $\mathcal{D} = \big\{\big(\boldsymbol{O}^{(i)}, \boldsymbol{U}^{(i)}\big)\big\}_{i=1}^N$. This approach is called identifiable VAE (iVAE). Most importantly, it can be proved that iVAE can identify latent variables $\boldsymbol{X}$ up to a permutation and pointwise transformation under the conditions stated in Theorem 2 of (Khemakhem et al., 2020).

---

[1]A brief description of variational autoencoders is given in Appendix A.

## 2.2 INVARIANT RISK MINIMIZATION

Arjovsky et al. (2019) introduced invariant risk minimization (IRM), whose goal is to construct an **invariant predictor** $f$ that performs well across all environments $\mathcal{E}_{all}$ by exploiting the varying degrees of spurious correlation naturally present in the training data collected from multiple environments $\mathcal{E}_{tr}$, where $\mathcal{E}_{tr} \subseteq \mathcal{E}_{all}$. Technically, they consider datasets $D_e := \{(\boldsymbol{o}_i^e, \boldsymbol{y}_i^e)\}_{i=1}^{n_e}$ from multiple training environments $e \in \mathcal{E}_{tr}$, where $\boldsymbol{o}_i^e \in \mathcal{O} \subseteq \mathbb{R}^d$ is the input observation and its corresponding label[2] is $\boldsymbol{y}_i^e \in \mathcal{Y} \subseteq \mathbb{R}^s$. The dataset $D_e$, collected from environment $e$, consists of examples identically and independently distributed according to some probability distribution $P(\boldsymbol{O}^e, \boldsymbol{Y}^e)$. The goal of IRM is to use these multiple datasets to learn a predictor $\boldsymbol{Y} = f(\boldsymbol{O})$ that achieves the minimum risk for all the environments. Here we define the risk reached by $f$ in environment $e$ as $R^e(f) = \mathbb{E}_{\boldsymbol{O}^e, \boldsymbol{Y}^e}[\ell(f(\boldsymbol{O}^e), \boldsymbol{Y}^e)]$. Then, the invariant predictor can be formally defined as below,

**Definition 1** (Invariant Predictor, Arjovsky et al. (2019)). *We say that a data representation $\Phi \in \mathcal{H}_\Phi : \mathcal{O} \to \mathcal{C}$ elicits an invariant predictor $w \circ \Phi$ across environments $\mathcal{E}$ if there is a classifier $w \in \mathcal{H}_w : \mathcal{C} \to \mathcal{Y}$ simultaneously optimal for all environments, that is, $w \in \arg\min_{\bar{w} \in \mathcal{H}_w} R^e(\bar{w} \circ \Phi)$ for all $e \in \mathcal{E}$.*

Mathematically, IRM can be phrased as the following constrained optimization problem:

$$\min_{\Phi \in \mathcal{H}_\Phi, w \in \mathcal{H}_w} \sum_{e \in \mathcal{E}_{tr}} R^e(w \circ \Phi) \quad \text{s.t. } w \in \arg\min_{\bar{w} \in \mathcal{H}_w} R^e(\bar{w} \circ \Phi), \forall e \in \mathcal{E}_{tr}. \tag{4}$$

Since this is a generally infeasible bi-leveled optimization problem, Arjovsky et al. (2019) rephrased it as a tractable penalized optimization problem by transfering the inner optimization routine to a penalty term. The main generalization result (Theorem 9 in Arjovsky et al. (2019)) states that if both $\Phi$ and $w$ come from the class of linear models (i.e., $\mathcal{H}_\Phi = \mathbb{R}^{n \times n}$ and $\mathcal{H}_w = \mathbb{R}^{n \times 1}$), under certain conditions on the diversity of training environments (Assumption 8 in Arjovsky et al. (2019)) and the data generation, the invariant predictor $w \circ \Phi$ across $\mathcal{E}_{tr}$ obtained by solving Eq. (4) remains invariant in $\mathcal{E}_{all}$. It is worth noting that Ahuja et al. (2020) reconsidered this IRM problem from the perspective of game theory, called IRMG for short. Although in the new formulation they proved that there exist such invariant predictors in $\mathcal{E}_{tr}$ when both $\Phi$ and $w$ are relaxed to the nonlinear models, their main generalization result in $\mathcal{E}_{all}$ holds only when both $\Phi$ and $w$ are linear models (Theorem 2 in Ahuja et al. (2020)).

## 3 PROBLEM SETUP

### 3.1 A MOTIVATING EXAMPLE

In this section, we extend the example which was introduced by Wright (1921) and discussed by Arjovsky et al. (2019), and provide a further in-depth analysis.

**Model 1.** *Consider the following structural equation model (SEM):*

$$X_1 \leftarrow Gaussian(0, \sigma_1(e)), \quad Y \leftarrow X_1 + Gaussian(0, \sigma_2(e)), \quad X_2 \leftarrow Y + Gaussian(0, \sigma_3(e)),$$

*where $\sigma_i(e) \geq 0$, varying in environment $e \in \mathcal{E}_{all}$, and $\mathcal{E}_{all}$ is the set of all environments.*

To ease exposition, here we consider the simple scenario in which $\mathcal{E}_{all}$ only contains all modifications varying the noises of $X_1$, $X_2$ and $Y$ within a finite range, i.e., $\sigma_i(e) \in [0, \sigma_{max}^2]$. Then, to predict $Y$ from $(X_1, X_2)$ using a least-square predictor $\hat{Y}^e = \hat{\alpha}_1 X_1^e + \hat{\alpha}_2 X_2^e$ for environment $e$, we can

- Case 1: regress from $X_1^e$, to obtain $\hat{\alpha}_1 = 1$ and $\hat{\alpha}_2 = 0$,
- Case 2: regress from $X_2^e$, to obtain $\hat{\alpha}_1 = 0$ and $\hat{\alpha}_2 = \frac{\sigma_1(e) + \sigma_2(e)}{\sigma_1(e) + \sigma_2(e) + \sigma_3(e)}$,
- Case 3: regress from $(X_1^e, X_2^e)$, to obtain $\hat{\alpha}_1 = \frac{\sigma_3(e)}{\sigma_2(e) + \sigma_3(e)}$ and $\hat{\alpha}_2 = \frac{\sigma_2(e)}{\sigma_2(e) + \sigma_3(e)}$.

In general scenarios (i.e., $\sigma_1(e) \neq 0$, $\sigma_2(e) \neq 0$, and $\sigma_3(e) \neq 0$), the regression using $X_1$ in Case 1 is an invariant correlation: this is the only regression whose coefficients do not vary with the environment $e$. By contrast, the regressions in both Case 2 and Case 3 have varying coefficients

---

[2]This setup for labels applies to both continuous and categorical data, where the categorical data can be encoded in the one-hot form.

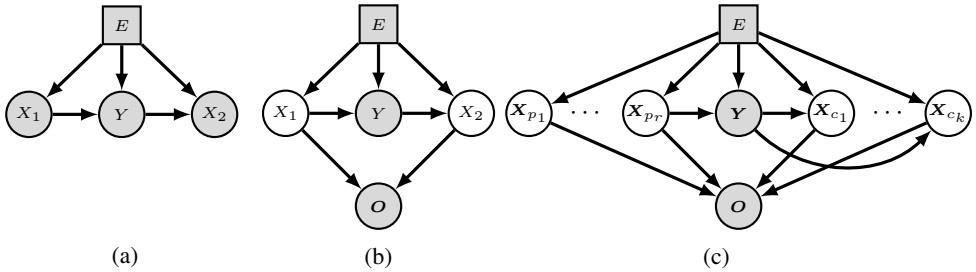

Figure 1: (a) Causal structure of Model 1. (b) A more practical extension of Model 1, where $X_1$ and $X_2$ are not directly observed and $O$ is their observation. (c) A general version of (b), where we assume there exist multiple unobserved variables. Each of them could be either a parent, a child of $Y$, or has no connection with $Y$. Grey nodes denote observed variables and white nodes represent unobserved variables.

depending on the environment $e$. Not surprisingly, only the invariant correlation in Case 1 would generalize well to new test environments.

From a practical perspective, let us take a closer look at Case 3. Because we do not know in advance that regressing from $X_1$ alone will lead to an invariant predictor, in practice we may do the regression from all the accessible data $(X_1^e, X_2^e)$. As aforementioned, when $\sigma_i(e) \neq 0$ for $i = 1, 2, 3$, the regression does not work. Actually any empirical risk minimization (ERM) algorithm purely minimizing training error (Vapnik, 1992) would not work in this setting. Invariant Causal Prediction (ICP) methods (Peters et al., 2015) also do not work, since the noise variance in $Y$ may change across environments. To this end, Arjovsky et al. (2019) proposed IRM. As aforementioned, however, IRM and IRMG can generalize well to unseen testing environments only in the linear setting. This motivates us to develop an approach to enabling the OOD generalization in the nonlinear setting (i.e., both $\Phi$ and $w$ are from the class of nonlinear models).

A more straightforward way to understand the motivating example is in its corresponding graphical representation[3], as shown in Fig. 1a. Following Peters et al. (2015), we treat the environment as a random variable $E$, where $E$ could be any information specific to the environment. For simplicity, we let $E$ be the environment index, i.e., $E \in \{1, \ldots, N\}$ and $N$ is the number of training environments. Note that, here we consider $E$ as a surrogate variable because it itself is not a causal variable (Zhang et al., 2017; Huang et al., 2020). From Fig. 1a, it is obvious to see that ICP does not work in this setting, since $Y \not\perp\!\!\!\perp E|X_1$. In fact, a more practical version appearing in real problems is present in Fig. 1b, where the true variables $\{X_1, X_2\}$ are unobserved and we only can observe their transformation $O$, which is a function of $\{X_1, X_2\}$. In this case, even if $Y$ is not affected by $E$ (i.e., remove the edge $E \rightarrow Y$), applying ICP to $O$ still does not work, since each variable (i.e., each dimension) of $O$ is jointly influenced by both $X_1$ and $X_2$. By contrast, both IRM and IRMG work when the transformation is linear, but not when it is nonlinear. These analyses are also empirically demonstrated in Section 5.1.

## 3.2 THE GENERAL SETTING

Inspired by the two dimensional example (i.e., $X = (X_1, X_2) \in \mathbb{R}^2$) described above, we naturally extend it to a more general multi-dimensional setting[4] as shown in Fig. 1c. Technically, we have $O \in \mathcal{O} \subseteq \mathbb{R}^d$, $Y \in \mathcal{Y} \subseteq \mathbb{R}^s$, $X = (X_{p_1}, \ldots, X_{p_r}, X_{c_1}, \ldots, X_{c_k}) \in \mathcal{X} \subseteq \mathbb{R}^{n(r+k)}$ (lower-dimensional, $n(r+k) \leq d$), where we assume each $X_i \in \mathbb{R}^n$ for simplicity. It is worth emphasising that except $X_{p_r}$, none of $\{X_{p_1}, \ldots, X_{p_{r-1}}\}$ has an arrow pointed to $Y$, meaning that all the parents of $Y$ are absorbed into $X_{p_r}$. Under this circumstance, more formally, we assume that

**Assumption 1.** $X_i \perp\!\!\!\perp X_j|Y, E$ for any $i \neq j$.

This assumption is not very restrictive but practically and theoretically reasonable enough to be able to cover various scenarios, due to the following reasons:

---

[3]The relation between SEM and its graphical representation is formally defined in Appendix B.

[4]As an initial work, we do not explicitly consider unobserved confounders in this paper. Precisely, for simplicity, we assume that there are no unobserved confounders between $X_i, Y, O$, and $E$. In fact, in some cases, our approach will not be affected even if there exist unobserved confounders. For example, if there were an unobserved confounder between $X_i$ and $O$, it would be absorbed into $X_i$ and would not affect our approach.

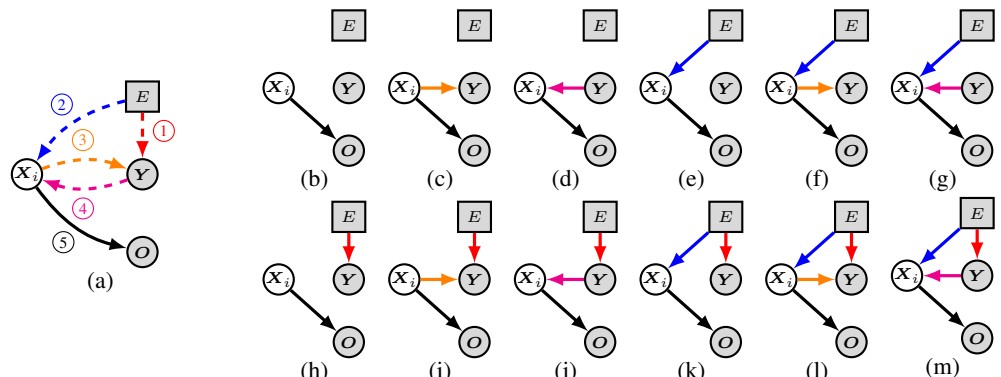

Figure 2: (a) General causal structure over $\{\boldsymbol{X}_i, \boldsymbol{Y}, \boldsymbol{O}, E\}$, where the arrow from $\boldsymbol{X}_i$ to $\boldsymbol{O}$ is a must-have connection and the other four might not be necessarily. (b) 12 possible causal structures from (a).

Firstly, Assumption 1 allows us to separately deal with each $\boldsymbol{X}_i$. When taking a closer look at each $\boldsymbol{X}_i$ in Fig. 1c, it is evident that there exist only five possible connections between $\boldsymbol{X}_i$, $\boldsymbol{Y}$, $E$, and $\boldsymbol{O}$, as shown in Fig. 2a. Among them, only the arrow from $\boldsymbol{X}_i$ to $\boldsymbol{O}$ must exist whilst the other four might not necessarily, which leads to 12 possible types of structures present in Figs. 2b-2m. These structures cover most scenarios in real-world applications, e.g., $\boldsymbol{X}_i$ could be either a parent or a child of $\boldsymbol{Y}$, be either affected or not by $E$, and even have no connection to $\boldsymbol{Y}$, etc. It is also worth noting that Assumption 1 does not rule out the possibility that there might exist certain correlations between all these latent variables.

Secondly, although it is well known that the key idea behind learning disentangled representations is that real-world data is generated by a few explanatory factors of variation, Locatello et al. (2019) show that unsupervised learning of disentangled representations is fundamentally impossible without inductive biases, and that well-disentangled models seemingly cannot be identified without supervision. They further suggest that disentanglement learning should be explicit about the role of inductive biases and supervision. In this sense, our Assumption 1 reasonably implements this idea.

Thirdly, as described in Section 2.1, identifiability of latent variables in iVAE requires a conditionally factorized prior distribution over the latent variables, as we did in Assumption 1, which is a key condition under which it is guaranteed that latent variables can be identified up to a permutation and pointwise transformation. This condition further inspires us to develop a generalization theory in the nonlinear setting that we will formulate in Section 4.3.

Now, under Assumption 1, solving Eq. (4) can be reduced to the key question of how to find the subset of $\boldsymbol{X}$ that are the direct causes of $\boldsymbol{Y}$, from the observational data $\{\boldsymbol{O}, E, \boldsymbol{Y}\}$, which will be discussed in the next section.

## 4  OUR APPROACH

In this section, we formally introduce our algorithm, namely Invariant Causal Representation Learning (ICRL), which consists of three phases and is summarized in Algorithm 1. The basic idea is that we first identify true latent variables by leveraging iVAE under Assumption 1 (Phase 1), then discover direct causes of $\boldsymbol{Y}$ (Phase 2), and finally learn an invariant predictor based on the identified direct causes (Phase 3).

### 4.1  PHASE 1: IDENTIFYING TRUE LATENT VARIABLES USING iVAE

Under Assumption 1, it is straightforward how to identify the true hidden factors $\boldsymbol{X}$ from $\boldsymbol{O}$ with the help of $\boldsymbol{Y}$ and $E$, by leveraging iVAE. We can directly substitute $\boldsymbol{U}$ with $(\boldsymbol{Y}, E)$ in Eqs. (1), and obtain its corresponding generative model:

$$p_{\boldsymbol{\theta}}(\boldsymbol{O}, \boldsymbol{X}|\boldsymbol{Y}, E) = p_{\boldsymbol{f}}(\boldsymbol{O}|\boldsymbol{X})p_{\boldsymbol{T}, \boldsymbol{\lambda}}(\boldsymbol{X}|\boldsymbol{Y}, E), \tag{5}$$
$$p_{\boldsymbol{f}}(\boldsymbol{O}|\boldsymbol{X}) = p_{\boldsymbol{\epsilon}}(\boldsymbol{O} - \boldsymbol{f}(\boldsymbol{X})). \tag{6}$$

---

**Algorithm 1:** Invariant Causal Representation Learning

---

**Phase 1:** We first learn the iVAE model, including the generative model and its corresponding inference model, by optimizing the evidence lower bound present in Eq. (8) on the data $\{O, Y, E\}$. Then, we use the learned iVAE model to infer the corresponding latent variable $X$ from $\{O, Y, E\}$, which is guaranteed to be identified up to a permutation and pointwise transformation.

**Phase 2:** Once obtaining $X$, according to Theorem 1, we can discover from them which are the direct causes Pa($Y$) of $Y$ by only performing **Rule 1.4**, **Rule 1.8**, **Rule 2.1**, and **Rule 3.1** described in Appendix E.3.

**Phase 3:** Once obtaining Pa($Y$), we can separately optimize Eq. (9) and Eq. (10) to learn the invariant data representation $\Phi$ and the invariant classifier $w$.

---

Likewise, we also obtain its corresponding prior distribution and lower bound:

$$p_{T,\lambda}(X|Y, E) = \prod_i \mathcal{Q}_i(X_i)/\mathcal{Z}_i(Y, E) \cdot \exp\left[\sum_{j=1}^{k} T_{i,j}(X_i)\lambda_{i,j}(Y, E)\right], \tag{7}$$

$$\mathcal{L}_{\text{phase1}}(\boldsymbol{\theta}, \boldsymbol{\phi}) := \mathbb{E}_{p_D}\left[\mathbb{E}_{q_\phi(X|O,Y,E)}\left[\log p_\theta(O, X|Y, E) - \log q_\phi(X|O, Y, E)\right]\right]. \tag{8}$$

This bound can be further expanded for computational convenience, which is given in Appendix C.

More importantly, under this setting we can directly borrow the identifiability result from Khemakhem et al. (2020) and then restate it below by replacing $U$ with $(Y, E)$.

**Theorem 1.** *Assume that we observe data sampled from a generative model defined according to Eqs. (5-7), with parameters $\boldsymbol{\theta} := (\boldsymbol{f}, \boldsymbol{T}, \boldsymbol{\lambda})$ and $k \geq 2$. Assume the following holds: (i) The set $\{O \in \mathcal{O}|\varphi_\epsilon(O) = 0\}$ has measure zero, where $\varphi_\epsilon$ is the characteristic function of the density $p_\epsilon$ defined in Eq. (6). (ii) The mixing function $\boldsymbol{f}$ in Eq. (6) is injective, and has all second order cross derivatives. (iii) The sufficient statistics $T_{i,j}$ in Eq. (7) are twice differentiable, and $(T_{i,j})_{1 \leq j \leq k}$ are linearly independent on any subset of $\mathcal{X}$ of measure greater than zero. (iv) There exist $nk + 1$ distinct points $(Y, E)^0, \ldots, (Y, E)^{nk}$ such that the matrix $L = \left(\boldsymbol{\lambda}((Y, E)^1) - \boldsymbol{\lambda}((Y, E)^0), \ldots, \boldsymbol{\lambda}((Y, E)^{nk}) - \boldsymbol{\lambda}((Y, E)^0)\right)$ of size $nk \times nk$ is invertible. Then the parameters $\boldsymbol{\theta}$ are identifiable up to a permutation and pointwise transformation.*

Theorem 1 deals with the general case $k \geq 2$, whose proof is given in Khemakhem et al. (2020).[5] According to Theorem 1, we can further have the following result of consistency of estimation.

**Theorem 2.** *Assume the following holds: (i) The family of distributions $q_\phi(X|O, Y, E)$ contains $p_\phi(X|O, Y, E)$. (ii) We maximize $\mathcal{L}_{\text{phase1}}(\boldsymbol{\theta}, \boldsymbol{\phi})$ with respect to both $\boldsymbol{\theta}$ and $\boldsymbol{\phi}$. Then in the limit of infinite data, iVAE learns the true parameters $\boldsymbol{\theta}^*$ up to a permutation and pointwise transformation.*

An immediate result of Theorem 1 and Theorem 2 is as follows,

**Theorem 3.** *Assume the hypotheses of Theorem 1 and Theorem 2 hold, then in the limit of infinite data, iVAE learns the true latent variables $X^*$ up to a permutation and pointwise transformation.*

The proofs of Theorem 2 and Theorem 3 are given in Appendix E. Theorem 3 says that we can leverage iVAE to learn the true conditionally factorized latent variables up to a permutation and pointwise transformation, which achieves our goal stated in Assumption 1.

## 4.2 PHASE 2: DISCOVERING DIRECT CAUSES

After identifying all the conditionally factorized latent variables $X$ from $O$, the question that comes is how to determine which component of $X := (X_{p_1}, \ldots, X_{p_r}, X_{c_1}, \ldots, X_{c_k})$ is the direct cause of $Y$, denoted by Pa($Y$). As discussed in Section 3.2, there are totally 12 possible types of structures over $\{X_i, Y, E, O\}$, as shown in Figs. 2b-2m. Apparently, only in the four of them (i.e., Figs. 2c, 2f, 2i, and 2l) does $X_i$ serve as a parent of $Y$. Note that, the reason that we also include Figs. 2i and 2l is that we follow Arjovsky et al. (2019) and consider a more general definition of invariance which allows for changes in the noise variance of $Y$. For convenience, we also put the definition in Appendix B. Surprisingly, given the data $\{X_i, Y, E, O\}$, we are able to distinguish all the 12

---

[5]We also provide a theorem dealing with the special case $k = 1$ in Appendix D.

structures by performing conditional independence tests (Spirtes et al., 2000; Zhang et al., 2012) and by leveraging causal discovery algorithms (Janzing et al., 2013; Peters et al., 2017; Zhang et al., 2017; Huang et al., 2020). This is summarized in Proposition 1 and the proof is given in Appendix E.

**Proposition 1.** *All the 12 structures shown in Figs. 2b-2m can be independently and completely distinguished in parallel.*

Proposition 1 allows us to efficiently discover all direct causes of $Y$ from $X$ by independently performing conditional independence tests and causal discovery algorithms for each $X_i$ in parallel. Besides, for each $X_i$ we only need to check if they are one of the four structures (i.e., Figs. 2c, 2f, 2i, and 2l) and this check can be also performed in parallel.

### 4.3 PHASE 3: LEARNING AN INVARIANT PREDICTOR

Once obtaining the invariant causal representation $\mathrm{Pa}(Y)$ for $Y$ across the training environments, learning an invariant predictor $w \circ \Phi$ in Eq. (4) is then reduced to two simpler independent optimization problems: (i) learning the invariant data representation $\Phi$ from $O$ to $\mathrm{Pa}(Y)$, and (ii) learning the invariant classifier $w$ from $\mathrm{Pa}(Y)$ to $Y$. Mathematically, these two optimization problems can be respectively phrased as

$$\min_{\Phi \in \mathcal{H}_\Phi} \sum\nolimits_{e \in \mathcal{E}_{tr}} R^e(\Phi) = \min_{\Phi \in \mathcal{H}_\Phi} \sum\nolimits_{e \in \mathcal{E}_{tr}} \mathbb{E}_{O^e, \mathrm{Pa}(Y^e)} \left[ \ell(\Phi(O^e), \mathrm{Pa}(Y^e)) \right], \tag{9}$$

$$\min_{w \in \mathcal{H}_w} \sum\nolimits_{e \in \mathcal{E}_{tr}} R^e(w) = \min_{w \in \mathcal{H}_w} \sum\nolimits_{e \in \mathcal{E}_{tr}} \mathbb{E}_{\mathrm{Pa}(Y^e), Y^e} \left[ \ell(w(\mathrm{Pa}(Y^e)), Y^e) \right]. \tag{10}$$

Eq. (9) and Eq. (10) guarantee that ICRL can achieve low error across $\mathcal{E}_{tr}$. Also, in Phase 1&2, we showed that ICRL can enforce invariance across $\mathcal{E}_{tr}$. Now this brings us to the question: how to enable the OOD generalization? In other words, how does ICRL achieve low error across $\mathcal{E}_{all}$? As Arjovsky et al. (2019) pointed out, low error across $\mathcal{E}_{tr}$ and invariance across $\mathcal{E}_{all}$ leads to low error across $\mathcal{E}_{all}$, because the generalization error of $w \circ \Phi$ respects standard error bounds once the data representation $\Phi$ eliciting an invariant predictor $w \circ \Phi$ across $\mathcal{E}_{all}$ is estimated. Thus, enabling the OOD generalization finally comes to the question: under which conditions does invariance across $\mathcal{E}_{tr}$ imply invariance across $\mathcal{E}_{all}$? Not surprisingly, $\mathcal{E}_{tr}$ must contain sufficient diversity to satisfy an underlying invariance across $\mathcal{E}_{all}$. Fortunately, the hypotheses of Theorem 1 automatically provides such a guarantee, and we therefore have the following result whose proof is in Appendix E.

**Proposition 2.** *Assume the hypotheses of Theorem 1 and Theorem 2 hold, then in the limit of infinite data, ICRL learns an invariant predictor $w \circ \Phi$ across $\mathcal{E}_{all}$.*

Proposition 2 tells us that under the assumptions given in Theorem 1 and Theorem 2, if ICRL can learn an invariant predictor $w \circ \Phi$ across $\mathcal{E}_{tr}$ in the limit of infinite data, then such a predictor $w \circ \Phi$ is invariant across $\mathcal{E}_{all}$.

## 5 EXPERIMENTS

We compare our approach with a variety of methods on both synthetic and real-world datasets. Due to space limit, we put in the supplement a detailed description of the datasets (Appendix F) and model architectures (Appendix H), as well as some in-depth analysis on experimental results (Appendix G). In all the comparisons, unless stated otherwise, we averaged the performance of the different methods over ten runs.

### 5.1 SYNTHETIC DATA

As a first experiment, in order to interpret how our approach works, we conduct a series of experiments on synthetic data generated according to an extension of the SEM in Model 1. This more practical extension is done by increasing the dimensionality of the two true features $X := (X_1, X_2)$ to 10 dimensions through a linear or nonlinear transformation, as illustrated in Fig. 1b. Technically, the goal is to predict $Y$ from $O$, where $O = g(X)$ and $g(\cdot)$ is called $X$ Transformer. We consider three types of transformations: (a) *Identity*: $g(\cdot)$ is the identity matrix $I \in \mathbb{R}^{2 \times 2}$, i.e.,

Table 1: Regression on synthetic data: Comparison of methods in terms of MSE (mean $\pm$ std deviation).

| X TRANSFORMER | ALGORITHM | TRAIN MSE ($\sigma_3 = \{0.2, 2\}$) | TEST MSE ($\sigma_3 = 5$) | TEST MSE ($\sigma_3 = 20$) | TEST MSE ($\sigma_3 = 100$) |
|---|---|---|---|---|---|
| Identity | ERM | $0.00 \pm 0.00$ | $0.00 \pm 0.00$ | $0.00 \pm 0.00$ | $0.00 \pm 0.00$ |
| | IRM | $0.00 \pm 0.00$ | $0.00 \pm 0.00$ | $0.00 \pm 0.00$ | $0.00 \pm 0.00$ |
| | F-IRM GAME | $0.81 \pm 0.37$ | $0.84 \pm 0.33$ | $1.18 \pm 0.32$ | $9.58 \pm 17.10$ |
| | V-IRM GAME | $0.80 \pm 0.24$ | $1.32 \pm 0.44$ | $9.66 \pm 10.86$ | $241.01 \pm 301.52$ |
| | **ICRL** | $0.01 \pm 0.03$ | $0.08 \pm 0.02$ | $0.45 \pm 0.06$ | $1.00 \pm 0.02$ |
| Linear | ERM | $0.00 \pm 0.00$ | $0.00 \pm 0.00$ | $0.00 \pm 0.00$ | $0.00 \pm 0.00$ |
| | IRM | $0.00 \pm 0.00$ | $0.00 \pm 0.00$ | $0.00 \pm 0.00$ | $0.00 \pm 0.00$ |
| | F-IRM GAME | $0.99 \pm 0.01$ | $1.00 \pm 0.01$ | $1.01 \pm 0.00$ | $1.11 \pm 0.16$ |
| | V-IRM GAME | $0.89 \pm 0.22$ | $1.69 \pm 1.46$ | $14.72 \pm 27.57$ | $380.88 \pm 759.94$ |
| | **ICRL** | $0.01 \pm 0.03$ | $0.03 \pm 0.01$ | $0.49 \pm 0.02$ | $1.02 \pm 0.03$ |
| Nonlinear | ERM | $0.06 \pm 0.01$ | $0.20 \pm 0.06$ | $6.56 \pm 6.01$ | $220.79 \pm 229.97$ |
| | IRM | $0.08 \pm 0.01$ | $0.20 \pm 0.04$ | $5.06 \pm 3.06$ | $149.60 \pm 104.85$ |
| | F-IRM GAME | $64.48 \pm 42.18$ | $954.42 \pm 634.18$ | $15606.28 \pm 10435.44$ | $360114.72 \pm 241206.47$ |
| | V-IRM GAME | $0.90 \pm 0.21$ | $1.74 \pm 0.34$ | $24.72 \pm 16.34$ | $780.67 \pm 729.68$ |
| | **ICRL** | $\mathbf{0.31 \pm 0.03}$ | $\mathbf{0.80 \pm 0.02}$ | $\mathbf{3.02 \pm 0.12}$ | $\mathbf{30.38 \pm 3.67}$ |

$O = g(X) = X$. (b) *Linear*: $g(\cdot)$ is a random matrix $S \in \mathbb{R}^{2 \times 10}$, i.e., $O = g(X) = X \cdot S$. (c) *Nonlinear*: $g(\cdot)$ is implemented by a multilayer perceptron with the 2 dimensional input and the 10 dimensional output, whose parameters are randomly initialized in advance. For simplicity, here we consider the regression task, in which the mean squared error (MSE) is used as a metric.

We consider an extremely simple scenario in which we fix $\sigma_1 = 1$ and $\sigma_2 = 0$ for all environments and only allow $\sigma_3$ to vary across environments. In this case, $\sigma_3$ controls how much the representation depends on the variable $X_2$, which is responsible for the spurious correlations. Each experiment draws 1000 samples from each of the five environments $\sigma_3 = \{0.2, 2, 5, 20, 100\}$, where the first two are for training and the rest for testing. We compare with several closely related baselines[6]: ERM, and two variants of IRMG: F-IRM Game (with $\Phi$ fixed to the identity) and V-IRM Game (with a variable $\Phi$).

As shown Table 1, in the cases of *Identity* and *Linear*, our approach is better than IRMG but only comparable with ERM and IRM. This might be because the identifiability result up to a pointwise nonlinear transformation renders the problem more difficult than itself, that is, converting the original identity or linear problem to a nonlinear problem. In the *Nonlinear* case, it is clear that the advantage of our approach becomes more obvious as the spurious correlation becomes stronger.

We also perform a series of experiments to further analyze our approach, including the analysis on the importance of Assumption 1 and on the necessity of iVAE in Phase 1, how accurately the direct causes can be recovered in Phase 2, and how well the two optimization problems can be addressed in Phase 3, all of which can be found in Appendix G.

## 5.2 COLORED MNIST AND COLORED FASHION MNIST

In this section, we conduct experiments on two datasets used in IRM and IRMG: Colored MNIST and Colored Fashion MNIST. We follow the same setting of Ahuja et al. (2020) to create these two datasets. The task is to predict a binary label assigned to each image which is originally grayscale but artificially colored in a way that correlated strongly but spuriously with the class label. We compare with 1) IRM, 2) two variants of IRMG: F-IRM Game (with $\Phi$ fixed to the identity) and V-IRM Game (with a variable $\Phi$), 3) three variants of ERM: ERM (on entire training data), ERM $e$ (on each environment $e$), and ERM GRAYSCALE (on data with no spurious correlations), 4) ROBUST MIN MAX (minimizing the maximum loss across the multiple environments). Table 2 shows that our approach significantly outperforms all the others on Colored Fashion MNIST. It is worth emphasising that both train and test accuracies of our method closely approach the ones of ERM GRAYSCALE and OPTIMAL, implying that it does approximately learn the true invariant causal representation with nearly no correlation with the color. We can achieve a similar conclusion from the results on Colored MNIST as shown in Table 3. However, this dataset is seemingly more difficult than the fashion version, because ERM GRAYSCALE still underperforms even though the spurious correlation with the color is removed, implying that it might involve some other spurious correlations. In this case, two training environments might be not enough to eliminate all the spurious correlations.

---

[6]We also tried ICP, but surprisingly ICP cannot find any parent of $Y$ even in the identity case.

Table 2: Colored Fashion MNIST: Comparison of methods in terms of accuracy (mean ± std deviation).

| ALGORITHM | TRAIN ACCURACY | TEST ACCURACY |
|---|---|---|
| ERM | $83.17 \pm 1.01$ | $22.46 \pm 0.68$ |
| ERM 1 | $81.33 \pm 1.35$ | $33.34 \pm 8.85$ |
| ERM 2 | $84.39 \pm 1.89$ | $13.16 \pm 0.82$ |
| ROBUST MIN MAX | $82.81 \pm 0.11$ | $29.22 \pm 8.56$ |
| F-IRM GAME | $62.31 \pm 2.35$ | $69.25 \pm 5.82$ |
| V-IRM GAME | $68.96 \pm 0.95$ | $70.19 \pm 1.47$ |
| IRM | $75.01 \pm 0.25$ | $55.25 \pm 12.42$ |
| **ICRL** | $\mathbf{74.32 \pm 0.43}$ | $\mathbf{73.14 \pm 0.56}$ |
| ERM GRAYSCALE | $74.79 \pm 0.37$ | $74.67 \pm 0.48$ |
| OPTIMAL | 75 | 75 |

Table 3: Colored MNIST: Comparison of methods in terms of accuracy (mean ± std deviation).

| ALGORITHM | TRAIN ACCURACY | TEST ACCURACY |
|---|---|---|
| ERM | $84.88 \pm 0.16$ | $10.45 \pm 0.66$ |
| ERM 1 | $84.84 \pm 0.21$ | $10.86 \pm 0.52$ |
| ERM 2 | $84.95 \pm 0.20$ | $10.05 \pm 0.23$ |
| ROBUST MIN MAX | $84.25 \pm 0.43$ | $15.24 \pm 2.45$ |
| F-IRM GAME | $63.37 \pm 1.14$ | $59.91 \pm 2.69$ |
| V-IRM GAME | $63.97 \pm 1.03$ | $49.06 \pm 3.43$ |
| IRM | $59.27 \pm 4.39$ | $62.75 \pm 9.59$ |
| **ICRL** | $\mathbf{70.34 \pm 0.29}$ | $\mathbf{66.21 \pm 1.42}$ |
| ERM GRAYSCALE | $71.81 \pm 0.47$ | $71.36 \pm 0.65$ |
| OPTIMAL | 75 | 75 |

# 6 RELATED WORK

**Invariant Prediction**   Peters et al. (2015) originally introduced the theory of Invariant Causal Prediction (ICP), aiming to find the *causal feature set* (i.e., all direct causes of a target variable of interest.) by exploiting the invariance property in causality which has been widely discussed under the term "autonomy", "modularity", and "stability" (Haavelmo, 1944; Aldrich, 1989; Hoover, 1990; Pearl, 2009; Dawid et al., 2010; Schölkopf et al., 2012). This invariance property assumed in ICP and its nonlinear extension (Heinze-Deml et al., 2018) is limited, because no intervention is allowed on the target variable $Y$. Besides, ICP methods implicitly assume that variables of interest $X$ are given. Magliacane et al. (2018) and Subbaswamy et al. (2019) attempt to find invariant predictors that maximally predictive using conditional independence tests and other graph-theoretic tools, both of which also assume that $X$ are given and further assume that additional information about the structure over $X$ is known. Arjovsky et al. (2019) reformulate this invariance as an optimization-based problem, allowing us to learn the invariant data representation from $O$ that is required to be linear transformations of $X$. Ahuja et al. (2020) extend IRM to the nonlinear setting from the perspective of game theory, but their nonlinear theory holds only in training environments.

**Domain Generalization**   Doman generalization emphasizes the ability to transfer acquired knowledge to domains unseen during training. A wide range of methods has been proposed for learning domain-invariant representations. Khosla et al. (2012) develop a max-margin classifier that explicitly exploits the effect of dataset bias and improve generalization ability to unseen domains. Fang et al. (2013) propose a metric learning approach based on structural SVM such that the neighbors of each training sample consist of examples from both the same and different domains. Muandet et al. (2013) propose a kernel-based optimization algorithm called Domain-Invariant Component Analysis (DICA), which aims to both minimize the discrepancy among domains and prevent the loss of relationship between input and output features. Ghifary et al. (2015) train a multi-task autoencoder that recognizes invariances among domains by learning to reconstruct analogs of original inputs from different domains. Motiian et al. (2017) learn an embedding subspace where samples from different domains are close if they have the same class labels, and far away if they bear different class labels. Li et al. (2018b) minimizes the differences in joint distributions to achieve target domain generalization through the application of a conditional invariant adversarial network. Li et al. (2018a) build on the adversarial autoencoders by considering maximum mean discrepancy regularization and aligning the domains' distributions.

# 7 CONCLUSION

We developed a novel framework to learn invariant predictors from a diverse set of training environments. This framework is based on a practical and general assumption: the data representation can be factorized when conditioning on the outcome and the environment. The assumption leads to a guarantee that the components in the representation can be identified up to a permutation and pointwise transformation. This allows us to further discover all the direct causes of the outcome, which enables generalization guarantees in the nonlinear setting. We hope our framework would inspire new ways to address the OOD generalization problem through the causal lens.

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

## A  VARIATIONAL AUTOENCODERS

We briefly describe the identifiable variational autoencoders (iVAEs) proposed by Khemakhem et al. (2020). As we know, the framework of variational autoencoders (VAEs) (Kingma & Welling, 2013; Rezende et al., 2014) allows us to efficiently learn deep latent-variable models and their corresponding inference models. Consider a simple latent variable model where $\boldsymbol{O} \in \mathbb{R}^d$ stands for an observed variable (random vector) and $\boldsymbol{X} \in \mathbb{R}^n$ for a latent variable. The VAE model actually learns a full generative model $p_{\boldsymbol{\theta}}(\boldsymbol{O}, \boldsymbol{X}) = p_{\boldsymbol{\theta}}(\boldsymbol{O}|\boldsymbol{X})p_{\boldsymbol{\theta}}(\boldsymbol{X})$ and an inference model $q_{\boldsymbol{\phi}}(\boldsymbol{X}|\boldsymbol{O})$ that approximates its posterior $p_{\boldsymbol{\theta}}(\boldsymbol{X}|\boldsymbol{O})$, where $\boldsymbol{\theta}$ is a vector of parameters of the generative model, $\boldsymbol{\phi}$ a vector of parameters of the inference model, and $p_{\boldsymbol{\theta}}(\boldsymbol{X})$ is a prior distribution over the latent variables. Instead of maximizing the data log-likelihood, we maximize its lower bound $\mathcal{L}_{\text{VAE}}(\boldsymbol{\theta}, \boldsymbol{\phi})$:

$$\log p_{\boldsymbol{\theta}}(\boldsymbol{O}) \geq \mathcal{L}_{\text{VAE}}(\boldsymbol{\theta}, \boldsymbol{\phi}) := \mathbb{E}_{q_{\boldsymbol{\phi}}(\boldsymbol{X}|\boldsymbol{O})}\left[\log p_{\boldsymbol{\theta}}(\boldsymbol{O}|\boldsymbol{X})\right] - \text{KL}\left(q_{\boldsymbol{\phi}}(\boldsymbol{X}|\boldsymbol{O})||p_{\boldsymbol{\theta}}(\boldsymbol{X})\right),$$

where we have used Jensen's inequality, and $\text{KL}(\cdot||\cdot)$ denotes the Kullback-Leibler divergence between two distributions.

## B  DEFINITIONS

For convenience, we restates some definitions here and please refer to the original papers (Arjovsky et al., 2019; Peters et al., 2017) for more details.

**Definition 2.** *A structural equation model (SEM) $\mathcal{C} := (\mathcal{S}, N)$ governing the random vector $\boldsymbol{X} = (X_1, \ldots, X_d)$ is a set of structural equations:*

$$\mathcal{S}_i : X_i \leftarrow f_i(\text{Pa}(X_i), N_i),$$

*where $\text{Pa}(X_i) \subseteq \{X_1, \ldots, X_d\} \setminus \{X_i\}$ are called the parents of $X_i$, and the $N_i$ are independent noise random variables. We say that "$X_i$ causes $X_j$" if $X_i \in \text{Pa}(X_j)$. We call causal graph of $\boldsymbol{X}$ to the graph obtained by drawing i) one node for each $X_i$, and ii) one edge from $X_i$ to $X_j$ if $X_i \in \text{Pa}(X_j)$. We assume acyclic causal graphs.*

**Definition 3.** *Consider a SEM $\mathcal{C} := (\mathcal{S}, N)$. An intervention $e$ on $\mathcal{C}$ consists of replacing one or several of its structural equations to obtain an intervened SEM $\mathcal{C}^e := (\mathcal{S}^e, N^e)$, with structural equations:*

$$\mathcal{S}_i^e : X_i^e \leftarrow f_i^e(\text{Pa}^e(X_i^e), N_i^e),$$

*The variable $\boldsymbol{X}^e$ is intervened if $\mathcal{S}_i \neq \mathcal{S}_i^e$ or $N_i \neq N_i^e$.*

**Definition 4.** *Consider a structural equation model (SEM) $\mathcal{S}$ governing the random vector $(X_1, \ldots, X_n, \boldsymbol{Y})$, and the learning goal of predicting $\boldsymbol{Y}$ from $\boldsymbol{X}$. Then, the set of all environments $\mathcal{E}_{all}(\mathcal{S})$ indexes all the interventional distributions $P(\boldsymbol{X}^e, \boldsymbol{Y}^e)$ obtainable by valid interventions $e$. An intervention $e \in \mathcal{E}_{all}(\mathcal{S})$ is valid as long as (i) the causal graph remains acyclic, (ii) $\mathbb{E}\left[\boldsymbol{Y}^e|\text{Pa}(\boldsymbol{Y})\right] = \mathbb{E}\left[\boldsymbol{Y}|\text{Pa}(\boldsymbol{Y})\right]$, and (iii) $\mathbb{V}\left[\boldsymbol{Y}^e|\text{Pa}(\boldsymbol{Y})\right]$ remains within a finite range.*

## C  DERIVATION

In Phase 1, the lower bound is defined by

$$
\begin{aligned}
\mathcal{L}_{\text{phase1}}(\boldsymbol{\theta}, \boldsymbol{\phi}) :=& \mathbb{E}_{p_D}\left[\mathbb{E}_{q_{\boldsymbol{\phi}}(\boldsymbol{X}|\boldsymbol{O}, \boldsymbol{Y}, E)}\left[\log p_{\boldsymbol{\theta}}(\boldsymbol{O}, \boldsymbol{X}|\boldsymbol{Y}, E) - \log q_{\boldsymbol{\phi}}(\boldsymbol{X}|\boldsymbol{O}, \boldsymbol{Y}, E)\right]\right] \\
=& \mathbb{E}_{p_D}\left[\mathbb{E}_{q_{\boldsymbol{\phi}}(\boldsymbol{X}|\boldsymbol{O}, \boldsymbol{Y}, E)}\left[\log p_{\boldsymbol{f}}(\boldsymbol{O}|\boldsymbol{X}) + \log p_{\boldsymbol{T}, \boldsymbol{\lambda}}(\boldsymbol{X}|\boldsymbol{Y}, E) - \log q_{\boldsymbol{\phi}}(\boldsymbol{X}|\boldsymbol{O}, \boldsymbol{Y}, E)\right]\right] \\
=& \mathbb{E}_{p_D}\left[\mathbb{E}_{q_{\boldsymbol{\phi}}(\boldsymbol{X}|\boldsymbol{O}, \boldsymbol{Y}, E)}\left[\log p_{\boldsymbol{f}}(\boldsymbol{O}|\boldsymbol{X})\right] + \mathbb{E}_{q_{\boldsymbol{\phi}}(\boldsymbol{X}|\boldsymbol{O}, \boldsymbol{Y}, E)}\left[\log p_{\boldsymbol{T}, \boldsymbol{\lambda}}(\boldsymbol{X}|\boldsymbol{Y}, E)\right]\right. \\
&\left. - \mathbb{E}_{q_{\boldsymbol{\phi}}(\boldsymbol{X}|\boldsymbol{O}, \boldsymbol{Y}, E)}\left[\log q_{\boldsymbol{\phi}}(\boldsymbol{X}|\boldsymbol{O}, \boldsymbol{Y}, E)\right]\right].
\end{aligned}
$$

The first term is the data log-likelihood and the third term has a closed-form solution,

$$\mathbb{E}_{q_{\boldsymbol{\phi}}(\boldsymbol{X}|\boldsymbol{O}, \boldsymbol{Y}, E)}\left[\log q_{\boldsymbol{\phi}}(\boldsymbol{X}|\boldsymbol{O}, \boldsymbol{Y}, E)\right] = -\frac{J}{2}\log(2\pi) + \frac{1}{2}\sum_{j=1}^{J}\left(1 + \log \sigma_j^2\right).$$

where $\sigma_j$ is simply denote the $j$-th element of the variational s.d. ($\sigma$) evaluated at datapoint $i$ that is simply a function of $(\boldsymbol{O}, \boldsymbol{Y}, E)$ and the variational parameters $\boldsymbol{\phi}$.

Now let us look at the second term,

$$\mathbb{E}_{q_\phi(\boldsymbol{X}|\boldsymbol{O},\boldsymbol{Y},E)}\left[\log p_{\boldsymbol{T},\boldsymbol{\lambda}}(\boldsymbol{X}|\boldsymbol{Y},E)\right]$$

$$=\mathbb{E}_{q_\phi(\boldsymbol{X}|\boldsymbol{O},\boldsymbol{Y},E)}\left[\log\left(\prod_i \frac{\mathcal{Q}_i(\boldsymbol{X}_i)}{\mathcal{Z}_i(\boldsymbol{Y},E)}\exp\left[\sum_{j=1}^{k}T_{i,j}(\boldsymbol{X}_i)\lambda_{i,j}(\boldsymbol{Y},E)\right]\right)\right]$$

$$=\mathbb{E}_{q_\phi(\boldsymbol{X}|\boldsymbol{O},\boldsymbol{Y},E)}\left[\sum_i \log\left(\frac{\mathcal{Q}_i(\boldsymbol{X}_i)}{\mathcal{Z}_i(\boldsymbol{Y},E)}\exp\left[\sum_{j=1}^{k}T_{i,j}(\boldsymbol{X}_i)\lambda_{i,j}(\boldsymbol{Y},E)\right]\right)\right]$$

$$\propto\mathbb{E}_{q_\phi(\boldsymbol{X}|\boldsymbol{O},\boldsymbol{Y},E)}\left[\sum_i\sum_{j=1}^{k}T_{i,j}(\boldsymbol{X}_i)\lambda_{i,j}(\boldsymbol{Y},E)\right]$$

$$\approx\frac{1}{L}\sum_l\sum_i\sum_{j=1}^{k}T_{i,j}(\boldsymbol{X}_i^l)\lambda_{i,j}(\boldsymbol{Y},E^l),$$

where we let the base measure $\mathcal{Q}_i(\boldsymbol{X}_i)=1$ and $L$ is the sample size.

## D  THEOREMS

**Theorem 4.** *Assume that we observe data sampled from a generative model defined according to Eqs. (5-7), with parameters $\boldsymbol{\theta}=(\boldsymbol{f},\boldsymbol{T},\boldsymbol{\lambda})$ and $k=1$. Assume the following holds: (i) The set $\{\boldsymbol{O}\in\mathcal{O}|\varphi_{\boldsymbol{\epsilon}}(\boldsymbol{O})=0\}$ has measure zero, where $\varphi_{\boldsymbol{\epsilon}}$ is the characteristic function of the density $p_{\boldsymbol{\epsilon}}$ defined in Eq. (6). (ii) The mixing function $\boldsymbol{f}$ in Eq. (6) is injective, and all partial derivatives of $\boldsymbol{f}$ are continuous. (iii) The sufficient statistics $T_{i,j}$ in Eq. (7) are differentiable almost everywhere and not monotonic, and $(T_{i,j})_{1\leq j\leq k}$ are linearly independent on any subset of $\mathcal{X}$ of measure greater than zero. (iv) There exist $nk+1$ distinct points $(\boldsymbol{Y},E)^0,\dots,(\boldsymbol{Y},E)^{nk}$ such that the matrix $L=\left(\boldsymbol{\lambda}((\boldsymbol{Y},E)^1)-\boldsymbol{\lambda}((\boldsymbol{Y},E)^0),\dots,\boldsymbol{\lambda}((\boldsymbol{Y},E)^{nk})-\boldsymbol{\lambda}((\boldsymbol{Y},E)^0)\right)$ of size $nk\times nk$ is invertible. Then the parameters $\boldsymbol{\theta}=(\boldsymbol{f},\boldsymbol{T},\boldsymbol{\lambda})$ are identifiable up to a permutation and pointwise transformation.*

## E  PROOFS

### E.1  PROOF OF THEOREM 2

This proof is similar to that of Theorem 4 in Khemakhem et al. (2020)

*Proof.* The loss function in Eq. 3 can be rephrased as follows:

$$\mathcal{L}_{\text{phase1}}(\boldsymbol{\theta},\boldsymbol{\phi})=\log p_{\boldsymbol{\theta}}(\boldsymbol{O}|\boldsymbol{Y},E)-KL\left(q_\phi(\boldsymbol{X}|\boldsymbol{O},\boldsymbol{Y},E)||p_{\boldsymbol{\theta}}(\boldsymbol{X}|\boldsymbol{O},\boldsymbol{Y},E)\right).$$

If the family of $q_\phi(\boldsymbol{X}|\boldsymbol{O},\boldsymbol{Y},E)$ is flexible enough to contain $p_{\boldsymbol{\theta}}(\boldsymbol{X}|\boldsymbol{O},\boldsymbol{Y},E)$, then by optimizing the loss over its parameter $\phi$, we will minimize the KL term which will eventually reach zero, and the loss will be equal to the log-likelihood. Under this circumstance, the iVAE inherits all the properties of maximum likelihood estimation. In this particular case, since our identifiability is guaranteed up to a permutation and pointwise transformation, the consistency of MLE means that we converge to the true parameter $\boldsymbol{\theta}^*$ up to a permutation and pointwise transformation in the limit of infinite data. Because true identifiability is one of the assumptions for MLE consistency, replacing it by identifiability up to a permutation and pointwise transformation does not change the proof but only the conclusion. □

### E.2  PROOF OF THEOREM 3

*Proof.* Theorem 1 and Theorem 2 guarantee that in the limit of infinite data, iVAE can learn the true parameters $\boldsymbol{\theta}^*:=(\boldsymbol{f}^*,\boldsymbol{T}^*,\boldsymbol{\lambda}^*)$ up to a permutation and pointwise transformation. Let $(\hat{\boldsymbol{f}},\hat{\boldsymbol{T}},\hat{\boldsymbol{\lambda}})$ be the parameters obtained by iVAE, and we therefore have $(\hat{\boldsymbol{f}},\hat{\boldsymbol{T}},\hat{\boldsymbol{\lambda}})\sim_P(\boldsymbol{f}^*,\boldsymbol{T}^*,\boldsymbol{\lambda}^*)$, where $\sim_P$ denotes the equivalence up to a permutation and pointwise transformation. If there were no

noise, this would mean that the learned $\hat{\boldsymbol{f}}$ transforms $\boldsymbol{O}$ into $\hat{\boldsymbol{X}} = \hat{\boldsymbol{f}}^{-1}(\boldsymbol{O})$ that are equal to $\boldsymbol{X}^* = (\boldsymbol{f}^*)^{-1}(\boldsymbol{O})$ up to a permutation and sighed scaling. If with noise, we obtain the posteriors of the latents up to an analogous indeterminacy. $\square$

### E.3 PROOF OF PROPOSITION 1

*Proof.* The following rules can be independently performed to distinguish all the 12 possible structures shown in Figs. 2b-2m. For clarity, we divide them into three groups.

**Group 1**    All the eight structures in this group can be discovered only by performing conditional independence tests.

- **Rule 1.1** If $\boldsymbol{X}_i \perp\!\!\!\perp \boldsymbol{Y}$, $\boldsymbol{X}_i \perp\!\!\!\perp E$, and $E \perp\!\!\!\perp \boldsymbol{Y}$, then Fig. 2b is discovered.

- **Rule 1.2** If $\boldsymbol{X}_i \perp\!\!\!\perp \boldsymbol{Y}$, $\boldsymbol{X}_i \perp\!\!\!\perp E$, and $E \not\!\perp\!\!\!\perp \boldsymbol{Y}$, then Fig. 2h is discovered.

- **Rule 1.3** If $\boldsymbol{X}_i \perp\!\!\!\perp \boldsymbol{Y}$, $\boldsymbol{X}_i \not\!\perp\!\!\!\perp E$, and $E \perp\!\!\!\perp \boldsymbol{Y}$, then Fig. 2e is discovered.

- **Rule 1.4** If $\boldsymbol{X}_i \not\!\perp\!\!\!\perp \boldsymbol{Y}$, $\boldsymbol{X}_i \perp\!\!\!\perp E$, and $E \not\!\perp\!\!\!\perp \boldsymbol{Y}$, then Fig. 2i is discovered.

- **Rule 1.5** If $\boldsymbol{X}_i \not\!\perp\!\!\!\perp \boldsymbol{Y}$, $\boldsymbol{X}_i \not\!\perp\!\!\!\perp E$, and $E \perp\!\!\!\perp \boldsymbol{Y}$, then Fig. 2g is discovered.

- **Rule 1.6** If $\boldsymbol{X}_i \not\!\perp\!\!\!\perp \boldsymbol{Y}$, $\boldsymbol{X}_i \not\!\perp\!\!\!\perp E$, $E \not\!\perp\!\!\!\perp \boldsymbol{Y}$, and $\boldsymbol{X}_i \perp\!\!\!\perp \boldsymbol{Y}|E$, then Fig. 2k is discovered.

- **Rule 1.7** If $\boldsymbol{X}_i \not\!\perp\!\!\!\perp \boldsymbol{Y}$, $\boldsymbol{X}_i \not\!\perp\!\!\!\perp E$, $E \not\!\perp\!\!\!\perp \boldsymbol{Y}$, and $\boldsymbol{X}_i \perp\!\!\!\perp E|\boldsymbol{Y}$, then Fig. 2j is discovered.

- **Rule 1.8** If $\boldsymbol{X}_i \not\!\perp\!\!\!\perp \boldsymbol{Y}$, $\boldsymbol{X}_i \not\!\perp\!\!\!\perp E$, $E \not\!\perp\!\!\!\perp \boldsymbol{Y}$, and $\boldsymbol{Y} \perp\!\!\!\perp E|\boldsymbol{X}_i$, then Fig. 2f is discovered.

**Group 2**    If $\boldsymbol{X}_i \not\!\perp\!\!\!\perp \boldsymbol{Y}$, $\boldsymbol{X}_i \perp\!\!\!\perp E$, and $E \perp\!\!\!\perp \boldsymbol{Y}$, then we can discover both Fig. 2c and Fig. 2d. These two structures cannot be further distinguished only by conditional independence tests, because they come from the same Markov equivalence class. Fortunately, we can further distinguish them by running binary causal discovery algorithms (Peters et al., 2017), e.g., ANM (Hoyer et al., 2009) or the bivariate fit model that is based on a best-fit criterion relying on a Gaussian Process regressor.

- **Rule 2.1** If $\boldsymbol{X}_i \not\!\perp\!\!\!\perp \boldsymbol{Y}$, $\boldsymbol{X}_i \perp\!\!\!\perp E$, and $E \perp\!\!\!\perp \boldsymbol{Y}$, and a chosen binary causal discovery algorithm prefers $\boldsymbol{X}_i \to \boldsymbol{Y}$ to $\boldsymbol{X}_i \leftarrow \boldsymbol{Y}$, then Fig. 2c is discovered.

- **Rule 2.2** If $\boldsymbol{X}_i \not\!\perp\!\!\!\perp \boldsymbol{Y}$, $\boldsymbol{X}_i \perp\!\!\!\perp E$, and $E \perp\!\!\!\perp \boldsymbol{Y}$, and a chosen binary causal discovery algorithm prefers $\boldsymbol{X}_i \leftarrow \boldsymbol{Y}$ to $\boldsymbol{X}_i \to \boldsymbol{Y}$, then Fig. 2d is discovered.

**Group 3**    If $\boldsymbol{X}_i \not\!\perp\!\!\!\perp \boldsymbol{Y}$, $\boldsymbol{X}_i \not\!\perp\!\!\!\perp E$, $E \not\!\perp\!\!\!\perp \boldsymbol{Y}$, $\boldsymbol{X}_i \not\!\perp\!\!\!\perp \boldsymbol{Y}|E$, $\boldsymbol{X}_i \not\!\perp\!\!\!\perp E|\boldsymbol{Y}$, and $\boldsymbol{Y} \not\!\perp\!\!\!\perp E|\boldsymbol{X}_i$, then we can discover both Fig. 2l and Fig. 2m. These two structures cannot be further distinguished only by conditional independence tests, because they come from the same Markov equivalence class. They also cannot be distinguished by any binary causal discovery algorithm, since both $\boldsymbol{X}_i$ and $\boldsymbol{Y}$ are affected by $E$. Fortunately, Zhang et al. (2017) provided a heuristic solution to this issue based on the invariance of causal mechanisms, i.e., $P(\text{cause})$ and $P(\text{effect}|\text{cause})$ change independently. The detailed description of their method is given in Section 4.2 of Zhang et al. (2017). For convenience, here we directly borrow their final result. Zhang et al. (2017) states that determining the causal direction between $\boldsymbol{X}_i$ and $\boldsymbol{Y}$ in Fig. 2l and Fig. 2m is finally reduced to calculating the following term:

$$\Delta_{\boldsymbol{X}_i \to \boldsymbol{Y}} = \left\langle \log \frac{\bar{P}(\boldsymbol{Y}|\boldsymbol{X}_i)}{\langle \hat{P}(\boldsymbol{Y}|\boldsymbol{X}_i)\rangle} \right\rangle, \tag{11}$$

where $\langle \cdot \rangle$ denotes the sample average, $\bar{P}(\boldsymbol{Y}|\boldsymbol{X}_i)$ is the empirical estimate of $P(\boldsymbol{Y}|\boldsymbol{X}_i)$ on all data points, and $\langle \hat{P}(\boldsymbol{Y}|\boldsymbol{X}_i)\rangle$ denotes the sample average of $\hat{P}(\boldsymbol{Y}|\boldsymbol{X}_i)$, which is the estimate of $P(\boldsymbol{Y}|\boldsymbol{X}_i)$ in each environment. We take the direction for which $\Delta$ is smaller to be the causal direction.

- **Rule 3.1** If $X_i \not\perp\!\!\!\perp Y$, $X_i \not\perp\!\!\!\perp E$, $E \not\perp\!\!\!\perp Y$, $X_i \not\perp\!\!\!\perp Y|E$, $X_i \not\perp\!\!\!\perp E|Y$, $Y \not\perp\!\!\!\perp E|X_i$, and $\Delta_{X_i \to Y}$ is smaller than $\Delta_{Y \to X_i}$, then Fig. 2l is discovered.

- **Rule 3.2** If $X_i \not\perp\!\!\!\perp Y$, $X_i \not\perp\!\!\!\perp E$, $E \not\perp\!\!\!\perp Y$, $X_i \not\perp\!\!\!\perp Y|E$, $X_i \not\perp\!\!\!\perp E|Y$, $Y \not\perp\!\!\!\perp E|X_i$, and $\Delta_{Y \to X_i}$ is smaller than $\Delta_{X_i \to Y}$, then Fig. 2m is discovered.

$\square$

### E.4 Proof of Proposition 2

*Proof.* Firstly, assumption (iii) and assumption (iv) in Theorem 1 are the requirements of the set of training environments containing sufficient diversity and satisfying an underlying invariance which holds across all the environments. Interestingly, assumption (iii) elicits Lemma 4 of Khemakhem et al. (2020), which is closely similar to the *linear general position* in Assumption 8 of Arjovsky et al. (2019). Thus, Lemma 4 can be similarly called the *nonlinear general position* in our generalization theory, whose proof can be found in Arjovsky et al. (2019). Secondly, when the set of training environments lie in this nonlinear general position and the other hypotheses of Theorem 1&2 hold, it is guaranteed in Theorem 3 that all the latent factors $X$ can be identified up to a permutation and pointwise transformation. Since this identifiability result holds under the assumptions guaranteeing that training environments contain sufficient diversity and satisfy an underlying invariance which holds across all the environments, it also holds across all the environments. Thirdly, Proposition 1 suggests that all the direct causes $Pa(Y)$ of $Y$ can be fully discovered, which also holds across all the environments due to the same reason above. Finally, the challenging bi-leveled optimization problem in both IRM and IRMG now can be reduced to two simpler independent optimization problems: (i) learning the invariant data representation $\Phi$ from $O$ to $Pa(Y)$, and (ii) learning the invariant classifier $w$ from $Pa(Y)$ to $Y$, as described in Eq. (9) and Eq. (10). For both (i) and (ii), since there exist no spurious correlations between $O$ and $Pa(Y)$ and between $Pa(Y)$ and $Y$, learning theory guarantees that in the limit of infinite data, we will converge to the true invariant data representation $\Phi$ and the true invariant classifier $w$. $\square$

It is worth noting that although assumption (iii) and assumption (iv) in Theorem 1 require complicated conditions to satisfy the diversity across training environments for generalization guarantees, it is not the case in practice. As we will observe in our experiments, it is often the case that two environments are sufficient to recover invariances.

## F Datasets

For convenience and completeness, we provide the descriptions of Colored MNIST Digits, Colored Fashion MNIST, and Office-Home here. Please refer to the original papers (Arjovsky et al., 2019; Ahuja et al., 2020; Gulrajani & Lopez-Paz, 2020; Venkateswara et al., 2017) for more details.

### F.1 Synthetic Data

For the nonlinear transformation, we use the MLP:

- Input layer: Input batch *(batch size, input dimension)*
- Layer 1: Fully connected layer, output size = 6, activation = ReLU
- Output layer: Fully connected layer, output size = 10

### F.2 Colored MNIST Digits

We use the exact same environment as in Arjovsky et al. (2019). Arjovsky et al. (2019) propose to create an environment for training to classify digits in MNIST digits data[7], where the images in MNIST are now colored in such a way that the colors spuriously correlate with the labels. The

---

[7]https://www.tensorflow.org/api_docs/python/tf/keras/datasets/mnist/load_data

task is to classify whether the digit is less than 5 (not including 5) or more than 5. There are three environments (two training containing 30,000 points each, one test containing 10,000 points) We add noise to the preliminary label ($\tilde{y} = 0$ if digit is between 0-4 and $\tilde{y} = 0$ if the digit is between 5-9) by flipping it with 25 percent probability to construct the final labels. We sample the color id $z$ by flipping the final labels with probability $p_e$, where $p_e$ is 0.2 in the first environment, 0.1 in the second environment, and 0.9 in the third environment. The third environment is the testing environment. We color the digit red if $z = 1$ or green if $z = 0$.

### F.3 COLORED FASHION MNIST

We modify the fashion MNIST dataset[8] in a manner similar to the MNIST digits dataset. Fashion MNIST data has images from different categories: "t-shirt", "trouser", "pullover", "dress", "coat", "sandal", "shirt", "sneaker", "bag", "ankle boots". We add colors to the images in such a way that the colors correlate with the labels. The task is to classify whether the image is that of foot wear or a clothing item. There are three environments (two training, one test) We add noise to the preliminary label ($\tilde{y} = 0$: "t-shirt", "trouser", "pullover", "dress", "coat", "shirt" and $\tilde{y} = 1$: "sandal", "sneaker", "ankle boots") by flipping it with 25 percent probability to construct the final label. We sample the color id $z$ by flipping the noisy label with probability $p_e$, where $p_e$ is 0.2 in the first environment, 0.1 in the second environment, and 0.9 in the third environment, which is the test environment. We color the object red if $z = 1$ or green if $z = 0$.

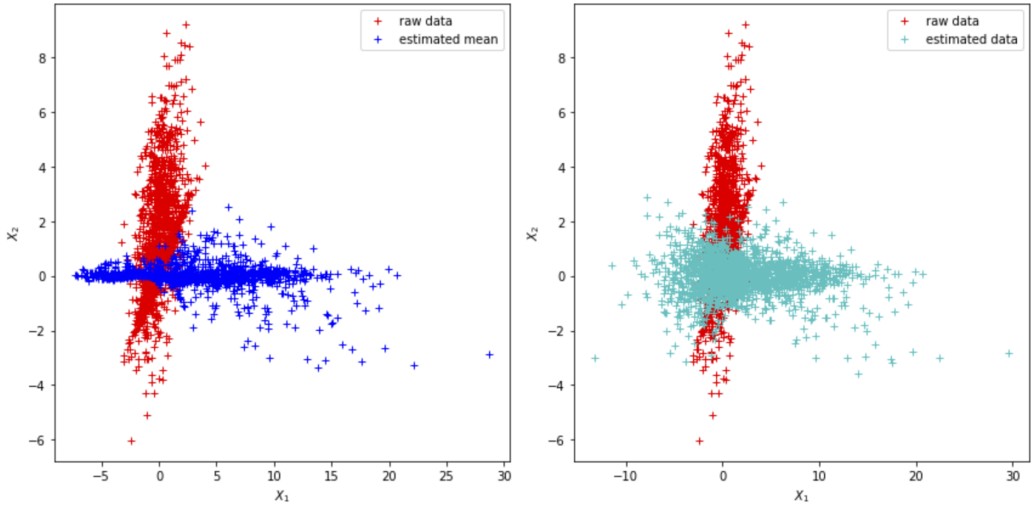

Figure 3: Left: Comparison of the raw data $X$ and the mean of $\hat{X}$ inferred through the learned inference model in the *Nonlinear* case. Right: Comparison of the raw data $X$ and the sampled points $\hat{X}$ using the reparameterization trick in the *Nonlinear* case. The comparisons clearly show that the inferred $\hat{X}$ is equal to $X$ up to a permutation and pointwise transformation.

## G   IN-DEPTH ANALYSIS ON SYNTHETIC DATA

### G.1   VERIFYING PHASE 1

Theorem 3 tells us that we can leverage iVAE to learn the true conditionally factorized latent variables up to a permutation and pointwise transformation. We empirically verify this point by comparing the raw data $X$ with the corresponding $\hat{X}$ inferred through the learned inference model in iVAE. Fig. 3 clearly shows that the inferred $\hat{X}$ is equal to $X$ up to a permutation and pointwise transformation.

---

[8]https://www.tensorflow.org/api_docs/python/tf/keras/datasets/fashion_mnist/load_data

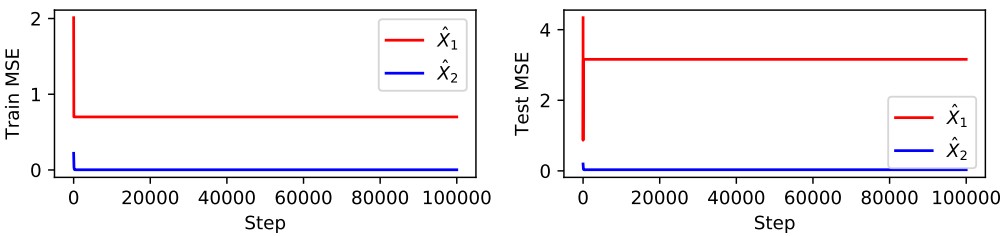

(a) Regression results in the *Linear* case in terms of MSE, where the inferred $\hat{X}_2$ is the identified cause.

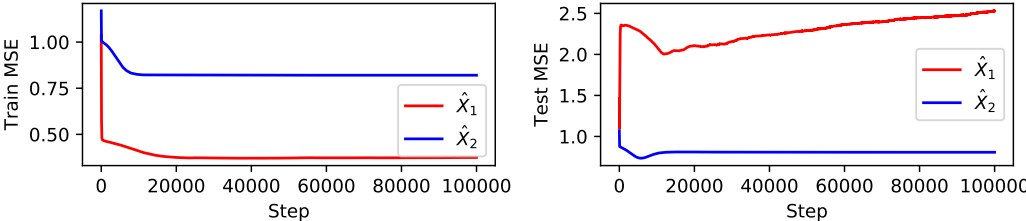

(b) Regression results in the *Nonlinear* case in terms of MSE, where the inferred $\hat{X}_2$ is the identified cause.

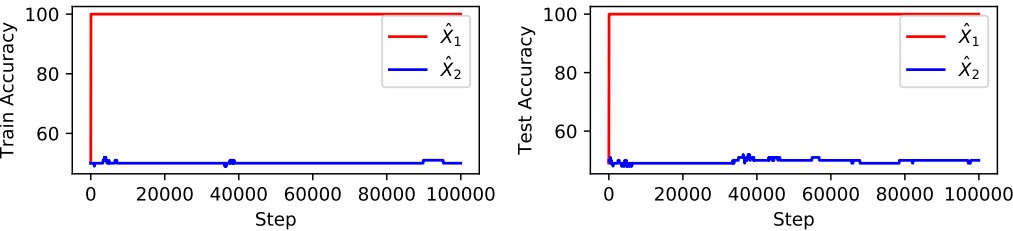

(c) Classification results on Synthetic Data in terms of accuracy, where the inferred $\hat{X}_1$ is the identified cause.

Figure 4: Comparison of the inferred $\hat{X}_1$ and $\hat{X}_2$ in terms of their final performance.

To show the importance of iVAE, we conduct an experiment in which we replace iVAE with the original VAE in Phase 1. As shown in Table 4, the performance of ICRL based on iVAE significantly outperforms the one based on VAE. It is worth noting that when VAE is instead used in Phase 1, it usually occurs in Phase 2 that either all the dimensions or no dimension of $\hat{X}$ are identified as the parents of $Y$. This is because all components of $\hat{X}$ are mixed together and will influence one another even when conditioning on $Y$ and $E$.

### G.2 VERIFYING PHASE 2

To show how well our method can identify the direct causes of $Y$ in Phase 2, we compare the final performance when the identified direct cause (i.e., $X_1$) and the identified non-cause (i.e., $X_2$) are respectively used in Phase 3 to learn the predictor. Note that, there might exist a permutation between the inferred $\{\hat{X}_1, \hat{X}_2\}$ and the true $\{X_1, X_2\}$. Fig. 4a and Fig. 4b show the results on the regression task, from which we can obviously see that the predictor elicited by the identified cause has a much better generalization performance. The classification result (i.e., $Y$ is binarized) in Fig. 4c further demonstrates this point.

### G.3 VERIFYING PHASE 3

In this experiment, we want to verify how well the data representation $\Phi$ can be learned by optimizing the loss in Eq. (9). The main idea is to check how well the learned $\hat{\Phi}$ can purely extract the

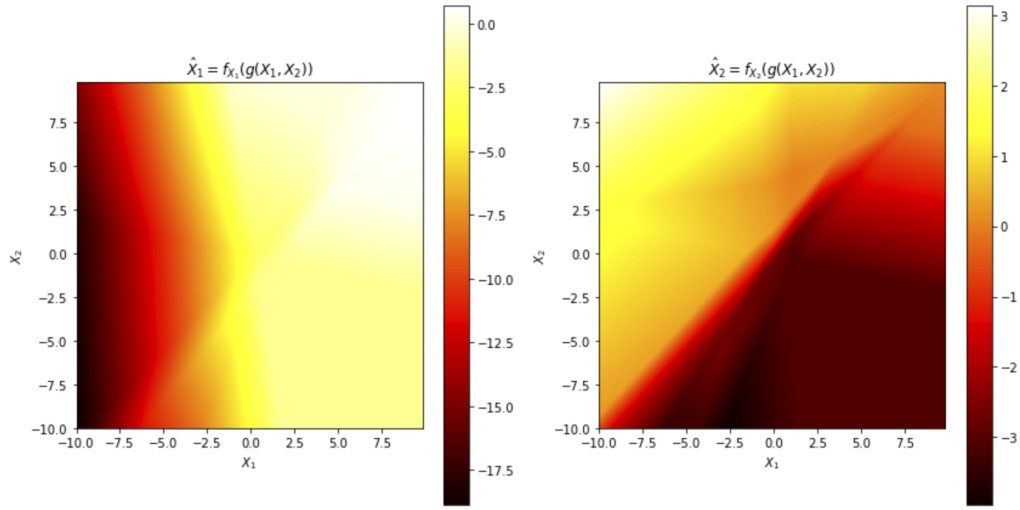

Figure 5: Left: Energy plot of $\hat{X}_1 = f_{X_1}(g(\boldsymbol{X}))$. Right: Energy plot of $\hat{X}_2 = f_{X_2}(g(\boldsymbol{X}))$. Note that, here $\hat{\Phi}_{X_i}$ is denoted by $f_{X_i}$.

Table 4: Results on synthetic Data: Comparison of iVAE and VAE used in Phase 1 in terms of MSE (mean $\pm$ std deviation).

| X TRANSFORMER | ALGORITHM | TRAIN MSE ($\sigma_3 = \{0.2, 2\}$) | TEST MSE ($\sigma_3 = 5$) | TEST MSE ($\sigma_3 = 20$) | TEST MSE ($\sigma_3 = 100$) |
|---|---|---|---|---|---|
| Nonlinear | ICRL-VAE | $0.26 \pm 0.09$ | $4.26 \pm 4.00$ | $42.03 \pm 48.67$ | $1174.96 \pm 1385.81$ |
| | **ICRL-iVAE** | $\mathbf{0.31 \pm 0.03}$ | $\mathbf{0.80 \pm 0.02}$ | $\mathbf{3.02 \pm 0.12}$ | $\mathbf{30.38 \pm 3.67}$ |

cause $X_1$ from $\boldsymbol{O}$. For this, we first learn $\hat{\Phi}_{X_i}$ for each $X_i$. Formally, we have

$$\hat{X}_i = \hat{\Phi}_{X_i}(\underbrace{g(\boldsymbol{X})}_{\boldsymbol{O}}), \text{ where } \boldsymbol{X} = (X_1, X_2).$$

Then, we observe how $\hat{X}_i$ will change while tuning $X_1$ and $X_2$ respectively. Fig. 5 shows the energy plots when $\hat{X}_1$ is the identified cause of $Y$ and $\hat{X}_2$ the child of $Y$. Note that, in the plots, $\hat{\Phi}_{X_i}$ is denoted by $f_{X_i}$. In theory, we are able to learn an invariant data representation $\Phi_{X_1}$ for $X_1$ from $\boldsymbol{O}$, because there is no spurious correlation between $X_1$ and $\boldsymbol{O}$. By contrast, we cannot learn an invariant data representation $\Phi_{X_2}$ for $X_2$ from $\boldsymbol{O}$, because there exist spurious correlations between $X_2$ and $\boldsymbol{O}$. The results shown in Fig. 5 clearly verify our theory. Specifically, in the left plot, $\hat{X}_1$ remains approximately unchanged when changing $X_2$ but it changes when changing $X_1$. However, in the right plot, $\hat{X}_2$ changes whether we change $X_1$ or $X_2$.

## H    MODEL ARCHITECTURES

In this section, we describe the architectures of different models used in different experiments.

### H.1    SYNTHETIC DATA

#### H.1.1    ERM

**Linear ERM**

- Input layer: Input batch *(batch size, input dimension)*
- Output layer: Fully connected layer, output size = 1

**Nonlinear ERM**

- Input layer: Input batch *(batch size, input dimension)*
- Layer 1: Fully connected layer, output size = 6, activation = ReLU
- Output layer: Fully connected layer, output size = 1

### H.1.2    IRM

**Linear Data Representation $\Phi$**

- Input layer: Input batch *(batch size, input dimension)*
- Output layer: Fully connected layer, output size = 1

**Nonlinear Data Representation $\Phi$**

- Input layer: Input batch *(batch size, input dimension)*
- Layer 1: Fully connected layer, output size = 6, activation = ReLU
- Output layer: Fully connected layer, output size = 1

### H.1.3    F-IRM GAME

**Linear Classifier $w$**

- Input layer: Input batch *(batch size, input dimension)*
- Output layer: Fully connected layer, output size = 1

**Nonlinear Classifier $w$**

- Input layer: Input batch *(batch size, input dimension)*
- Layer 1: Fully connected layer, output size = 6, activation = ReLU
- Output layer: Fully connected layer, output size = 1

### H.1.4    V-IRM GAME

**Linear Data Representation $\Phi$**

- Input layer: Input batch *(batch size, input dimension)*
- Output layer: Fully connected layer, output size = 2

**Nonlinear Data Representation $\Phi$**

- Input layer: Input batch *(batch size, input dimension)*
- Layer 1: Fully connected layer, output size = 6, activation = ReLU
- Output layer: Fully connected layer, output size = 2

**Linear Classifier $w$**

- Input layer: Input batch *(batch size, 2)*
- Output layer: Fully connected layer, output size = 1

**Nonlinear Classifier $w$**

- Input layer: Input batch *(batch size, 2)*
- Layer 1: Fully connected layer, output size = 6, activation = ReLU
- Output layer: Fully connected layer, output size = 1

### H.1.5 ICRL

**iVAE Linear Prior**

- Input layer: Input batch *(batch size, input dimension)*
- Mean Output layer: $\mathbf{0}$, which is a vector full of $0$ with the length 2
- Log Variance Output layer: Fully connected layer, output size = 2

**iVAE Nonlinear Prior**

- Input layer: Input batch *(batch size, input dimension)*
- Layer 1: Fully connected layer, output size = 6, activation = ReLU
- Mean Output layer: $\mathbf{0}$, which is a vector full of $0$ with the length 2
- Log Variance Output layer: Fully connected layer, output size = 2

**iVAE Linear Encoder**

- Input layer: Input batch *(batch size, input dimension)*
- Mean Output layer: Fully connected layer, output size = 2
- Log Variance Output layer: Fully connected layer, output size = 2

**iVAE Nonlinear Encoder**

- Input layer: Input batch *(batch size, input dimension)*
- Layer 1: Fully connected layer, output size = 6, activation = ReLU
- Mean Output layer: Fully connected layer, output size = 2
- Log Variance Output layer: Fully connected layer, output size = 2

**iVAE Linear Decoder**

- Input layer: Input batch *(batch size, 2)*
- Mean Output layer: Fully connected layer, output size = output dimension
- Variance Output layer: $0.01 \times \mathbf{1}$, where $\mathbf{1}$ is a vector full of $1$ with the length of output dimension

**iVAE Nonlinear Decoder**

- Input layer: Input batch *(batch size, 2)*
- Layer 1: Fully connected layer, output size = 6, activation = ReLU
- Mean Output layer: Fully connected layer, output size = output dimension
- Variance Output layer: $0.01 \times \mathbf{1}$, where $\mathbf{1}$ is a vector full of $1$ with the length of output dimension

**Linear Data Representation $\Phi$**

- Input layer: Input batch *(batch size, input dimension)*
- Output layer: Fully connected layer, output size = 1

**Nonlinear Data Representation $\Phi$**

- Input layer: Input batch *(batch size, input dimension)*
- Layer 1: Fully connected layer, output size = 6, activation = ReLU
- Output layer: Fully connected layer, output size = 1

**Linear Classifier** $w$

- Input layer: Input batch *(batch size, 1)*
- Output layer: Fully connected layer, output size = 1

**Nonlinear Classifier** $w$

- Input layer: Input batch *(batch size, 1)*
- Layer 1: Fully connected layer, output size = 6, activation = ReLU
- Output layer: Fully connected layer, output size = 1

## H.2 Colored MNIST Digits and Colored Fashion MNIST

**iVAE Prior**

- Input layer: Input batch *(batch size, input dimension)*
- Layer 1: Fully connected layer, output size = 100, activation = ReLU
- Mean Output layer: $\mathbf{0}$, which is a vector full of $0$ with the length 100
- Log Variance Output layer: Fully connected layer, output size = 100

**iVAE $O$-Encoder**

- Input layer: Input batch *(batch size, 2, 28, 28)*
- Layer 1: Convolutional layer, output channels = 32, kernel size = 3, stride = 2, padding = 1, activation = ReLU
- Layer 2: Convolutional layer, output channels = 32, kernel size = 3, stride = 2, padding = 1, activation = ReLU
- Layer 3: Convolutional layer, output channels = 32, kernel size = 3, stride = 2, padding = 1, activation = ReLU
- Output layer: Flatten

**iVAE $(Y, E)$-Encoder**

- Input layer: Input batch *(batch size, input dimension)*
- Output layer: Fully connected layer, output size = 100, activation = ReLU

**iVAE $(O, Y, E)$-Merger/Encoder**

- Input layer: Input batch *(batch size, input dimension)*
- Layer 1: Fully connected layer, output size = 100, activation = ReLU
- Mean Output layer: Fully connected layer, output size = 100
- Log Variance Output layer: Fully connected layer, output size = 100

**iVAE Decoder**

- Input layer: Input batch *(batch size, 100)*
- Layer 1: Fully connected layer, output size = $32 \times 4 \times 4$, activation = ReLU
- Layer 2: Reshape to *(batch size, 32, 4, 4)*
- Layer 3: Deconvolutional layer, output channels = 32, kernel size = 3, stride = 2, padding = 1, outpadding = 0, activation = ReLU
- Layer 4: Deconvolutional layer, output channels = 32, kernel size = 3, stride = 2, padding = 1, outpadding = 1, activation = ReLU

- Layer 5: Deconvolutional layer, output channels = 2, kernel size = 3, stride = 2, padding = 1, outpadding = 1
- Mean Output layer: activation = Sigmoid
- Variance Output layer: $0.01 \times \mathbf{1}$, where $\mathbf{1}$ is a matrix full of $1$ with the size of $2 \times 28 \times 28$.

**Data Representation** $\Phi$

- Input layer: Input batch *(batch size, 2, 28, 28)*
- Layer 1: Convolutional layer, output channels = 32, kernel size = 3, stride = 2, padding = 1, activation = ReLU
- Layer 2: Convolutional layer, output channels = 32, kernel size = 3, stride = 2, padding = 1, activation = ReLU
- Layer 3: Convolutional layer, output channels = 32, kernel size = 3, stride = 2, padding = 1, activation = ReLU
- Layer 4: Flatten
- Mean Output layer: Fully connected layer, output size = 100
- Log Variance Output layer: Fully connected layer, output size = 100

**Classifier** $w$

- Input layer: Input batch *(batch size, 100)*
- Layer 1: Fully connected layer, output size = 100, activation = ReLU
- Output layer: Fully connected layer, output size = 1, activation = Sigmoid

