# OpenReview forum: "Invariant Causal Representation Learning"
_ICLR.cc/2021/Conference — Reject_

### Official Review · AnonReviewer4 · 2020-10-25
**Proposes some interesting ideas, but the novelty is limited and experimental results are unconvincing**

**Rating:** 5
**Confidence:** 4

**Review:**

The paper proposes Invariant Causal Representation Learning, which seeks to learn representations for downstream tasks that are based on only causally invariant latent variables so the representation is robust to shifts in the test environment.

A model is assumed where an environment variable is a cause of all the latent variables and a target. The iVAE algorithm is used to learn the latent variable model. Then, a series of conditional independence tests and bivariate causal discovery methods are used to distinguish which latent variables correspond to causes (effects) of the target. Finally representations are learned from the observed variables to the causal latent variables of the target and then from these variables to the target.

The approach is evaluated using synthetic data and semi-synthetic data based on MNIST.

The clarity and organization of the paper could be improved. The algorithm should be moved from the appendix to the main text and the procedure should be described more holistically to give the reader an outline before diving into the details of each component section. The experiments section is also very unclear.

The main issue that remains unclear to me is how the environment variable E is being used explicitly. It doesn’t seem clear to me that you would generally have access to E, but it is required in the rules necessary to determine the direct causes of Y. What are the E variables being used in each of the experiments?

The novelty seems somewhat limited. It seems the theoretical results can be divided into (a) results about identifiability of the latent variable model and (b) the method for identifying the direct causes. Some of (a) follows directly from Khemakhem et al. (2020) - it is difficult to determine whether there is sufficient novelty here. (b) follows directly from well known constraint-based and bivariate causal discovery approaches.

The experiments are unconvincing. The proposed method outperforms existing approaches in a high noise synthetic data example and a kind of adversarial example where grayscale MNIST is colored in a way that is strongly correlated with the class label (the experimental setup and evaluation metric is very confusing here). It would be more convincing to the proposed method evaluated in a more realistic setting.

Further, there are no (synthetic) experiments which confirm that the proposed method does in fact learn the causally relevant latent variables and its robustness in doing so. Since identifying the correct causal latent variables requires multiple conditional independence tests and bivariate causal discovery methods (on latent variables which may be estimated incorrectly), there is an obvious concern about how robust this procedure is in practice. It would be more convincing to see (e.g.) precision and recall with regard to selecting the correct causal latent variables when the ground truth is known.

In summary, the paper introducing some interesting ideas, but the clarity could be improved, the novelty may be somewhat limited and the experimental results could be improved.

---

> ### Author Response · Authors · 2020-11-24
> **Response to Reviewer 4**
>
> Thank you for your comments.
>
> **Question 1.**
>
> "*The clarity and organization of the paper could be improved. The algorithm should be moved from the appendix to the main text and the procedure should be described more holistically to give the reader an outline before diving into the details of each component section. The experiments section is also very unclear.*"
>
> **Authors' Response**:
>
> Thank you for the suggestion. We have updated all in the revision.
>
> **Question 2.**
>
> "*The main issue that remains unclear to me is how the environment variable E is being used explicitly. It doesn’t seem clear to me that you would generally have access to E, but it is required in the rules necessary to determine the direct causes of Y. What are the E variables being used in each of the experiments?*"
>
> **Authors' Response**:
>
> In this paper, the environment variable $E$ is only an environment or domain index. For example, if we have $N$ training environments, then the environment variable $E$ takes value in {1, $\ldots$, N}. We have clarified it in the last paragraph of Section 3.1.
>
> **Question 3.**
>
> "*The novelty seems somewhat limited. It seems the theoretical results can be divided into (a) results about identifiability of the latent variable model and (b) the method for identifying the direct causes. Some of (a) follows directly from Khemakhem et al. (2020) - it is difficult to determine whether there is sufficient novelty here. (b) follows directly from well known constraint-based and bivariate causal discovery approaches.*"
>
> **Authors' Response**:
>
> It is worth emphasizing that our contribution in this paper is to propose **a novel learning paradigm** that enables OOD generalization **in the nonlinear setting**. This challenging problem, which was not solved before, is decently addressed in our paper by creatively integrating some existing methods in a comprehensive manner. Empirical results also demonstrate that our approach significantly outperforms IRM and IRMG in the nonlinear setting. Hence, our work would be a complement to the community of OOD generalization.
>
> **Question 4.**
>
> "*The experiments are unconvincing. The proposed method outperforms existing approaches in a high noise synthetic data example and a kind of adversarial example where grayscale MNIST is colored in a way that is strongly correlated with the class label (the experimental setup and evaluation metric is very confusing here). It would be more convincing to the proposed method evaluated in a more realistic setting.*"
>
> **Authors' Response**:
>
> In this paper, we followed the same experiment settings of the pioneering works on the OOD generalization (i.e., IRM and IRMG) to conduct all the experiments for the fair comparison. The main goal of our experiments in the paper is to **CONCEPTUALLY** demonstrate that the proposed method can enable the OOD generalization in the nonlinear setting.
>
> **Question 5.**
>
> "*Further, there are no (synthetic) experiments which confirm that the proposed method does in fact learn the causally relevant latent variables and its robustness in doing so. Since identifying the correct causal latent variables requires multiple conditional independence tests and bivariate causal discovery methods (on latent variables which may be estimated incorrectly), there is an obvious concern about how robust this procedure is in practice. It would be more convincing to see (e.g.) precision and recall with regard to selecting the correct causal latent variables when the ground truth is known.*"
>
> **Authors' Response**:
>
> Considering that the ground truth of the causal latent variables in the image experiments is unknown, we also conducted the experiments on the fully synthetic data in Section 5.1. In Appendix G, we provide an in-depth analysis on our approach, including the analysis on the importance of Assumption 1 and on the necessity of iVAE in Phase 1, how accurately and robustly the direct causes can be recovered in Phase 2, and how well the two optimization problems can be addressed in Phase 3.

---

### Official Review · AnonReviewer2 · 2020-10-26
**Interesting topic but the assumption and the experiments are not convincing.**

**Rating:** 4
**Confidence:** 4

**Review:**

This paper proposes an invariant causal representation learning paradigm in the nonlinear setting. Based on a conditional factorial assumption, they proved identifiability up to a linear transform. The ICRL objective, in this case, is able to discover all the direct causes of the outcome, and thus enables OOD generalization.

The novelty of the paper seems to be in the generalization of the IRM framework to the nonlinear case which is interesting to me.
The authors combined iVAE and IRM to solve this problem. Overall the paper is clearly written and easy to follow, but some conceptual issues remain.

Here are my issues with the paper:
- How to verify assumption 1 in a real case? Although the authors argued this assumption is not very restrictive and similar to the assumption in the iVAE paper, I think there is a difference between these two papers. In the iVAE paper, they assumed the latent variables to be conditionally factorial, while here the authors assume the potential causals (unobserved variables) are independent.

- After discovering direct causes, they still need the IRM phase to learn an invariant predictor. IRM itself can identify spurious causes and learn an invariant predictor, so what is the gain of learning the first two phases? What if some spurious causations are wrongly detected by the second phase, will it affect the predictor?

-The synthetic data experiment is not convincing at all. ICRL outperforms ERM and IRM in a very extreme case, where all the algorithms perform terribly, I don't think I can conclude ICRL is a better algorithm among others from this test case. If the authors can visually show the invariant representation of ICRL is more robust, that would be a good illustration.

- In the colored MNIST experiment, I assume the setting is the same as the IRM paper. While they said their IRM can reach 66.9+-2.5 acc, which is 7 percent higher than this paper and even higher than ICRL. So I wonder what causes this gap.

---

> ### Author Response · Authors · 2020-11-24
> **Response to Reviewer 2**
>
> Thank you for your feedback.
>
> **Question 1.**
>
> "*How to verify assumption 1 in a real case? Although the authors argued this assumption is not very restrictive and similar to the assumption in the iVAE paper, I think there is a difference between these two papers. In the iVAE paper, they assumed the latent variables to be conditionally factorial, while here the authors assume the potential causals (unobserved variables) are independent.*"
>
> **Authors' Response**:
>
> In fact, following the iVAE paper, in this paper we also assume the latent variables to be conditionally factorial, which is formally stated in Assumption 1.
>
> Note that, like many other areas (e.g., healthcare, epidemiology, medicine, etc.) in causality, the only way to verify the assumed causal diagram is through experiment. Empirical results demonstrate that this assumption works quite well.
>
> **Question 2.**
>
> "*After discovering direct causes, they still need the IRM phase to learn an invariant predictor. IRM itself can identify spurious causes and learn an invariant predictor, so what is the gain of learning the first two phases? What if some spurious causations are wrongly detected by the second phase, will it affect the predictor?*"
>
> **Authors' Response**:
>
> Compared to IRM, our method has at least two advantages.
>
> First, the challenging bi-leveled optimization problem in IRM can be reduced to two simpler independent optimization problems: (i) learning the invariant data representation $\Phi$ from $O$ to $\text{Pa}(Y)$, and (ii) learning the invariant classifier $w$ from $\text{Pa}(Y)$ to $Y$. Both (i) and (ii) can be separately performed in a more efficient and effective manner.
>
> Second, our method has generalization guarantees in the nonlinear setting, whist IRM only works in the linear setting. This guarantee come from the basic idea that for both (i) and (ii), since there exist no spurious correlations between ${O}$ and $\text{Pa}({Y})$ and between $\text{Pa}({Y})$ and ${Y}$, learning theory guarantees that in the limit of infinite data, we will converge to the true invariant data representation $\Phi$ and the true invariant classifier $w$.
>
> If some spurious causations are wrongly detected by the second phase, it will affect the predictor for sure.
>
> **Question 3.**
>
> "*The synthetic data experiment is not convincing at all. ICRL outperforms ERM and IRM in a very extreme case, where all the algorithms perform terribly, I don't think I can conclude ICRL is a better algorithm among others from this test case. If the authors can visually show the invariant representation of ICRL is more robust, that would be a good illustration.*"
>
> **Authors' Response**:
>
> Considering that the ground truth of the causal latent variables in the image experiments is unknown, we also conducted the experiments on the fully synthetic data in Section 5.1. In Appendix G, we provide an in-depth analysis on our approach, including the analysis on the importance of Assumption 1 and on the necessity of iVAE in Phase 1, how accurately and robustly the direct causes can be recovered in Phase 2, and how well the two optimization problems can be addressed in Phase 3.
>
> **Question 4.**
>
> "*In the colored MNIST experiment, I assume the setting is the same as the IRM paper. While they said their IRM can reach 66.9+-2.5 acc, which is 7 percent higher than this paper and even higher than ICRL. So I wonder what causes this gap.*"
>
> **Authors' Response**:
>
> Since the IRMG paper includes more baselines, for a fair comparison we followed their experimental setting and directly used their dataset. The baseline results in our paper directly come from the IRMG paper. The gap might be caused by the preprocessing methods used in the IRMG paper while creating the datasets.

---

### Official Review · AnonReviewer3 · 2020-10-28
**Interesting approach to an important problem, but with a substantial flaw**

**Rating:** 4
**Confidence:** 4

**Review:**

This paper proposes a method for learning invariant (nonlinear) data representations and classifiers, using data from multiple domains. A key step in the method is to discover the direct causes of the outcome of interest from a set of latent variables that are recovered from observed variables via identifiable VAE. The problem being tackled is significant, and the general idea is interesting and sensible. The empirical results also look encouraging. However, there appears to be a major flaw in the theoretical setup. In order to apply identifiable VAE, the method needs to assume that any two latent variables are independent conditional on the outcome variable and the domain index (Assumption 1). But what about latent variables that are direct causes of the outcome variable? If both X_1 and X_2 are direct causes of Y, generically they will not be independent conditional on Y and E, will they? In the motivating example, only one latent variable is a direct cause of the outcome, so this issue does not arise, but the ambition, as I understand it, is to handle any number of direct causes. I don't see how Assumption 1 can be justified as a plausible assumption in the general case where more than one latent variable is a direct cause of the outcome.

Moreover, Theorem 4 is less than fully rigorous and is misleading. For example, the proof of Theorem 4 invokes a method for inferring causal directions in Zhang et al. (2017), but as far as I know, that method does not yet have a rigorous theoretical justification. As it is formulated, Theorem 4 sounds like a theoretical identifiability result, and as such is not rigorously established by the proof given in the paper.

---

> ### Author Response · Authors · 2020-11-24
> **Response to Reviewer 3**
>
> Thank you for your comments.
>
> **Question 1**:
>
> "*I don't see how Assumption 1 can be justified as a plausible assumption in the general case where more than one latent variable is a direct cause of the outcome.*"
>
> **Authors' Response**:
>
> In fact, Assumption 1 applies to the general case where more than one latent variable is a direct cause of the outcome. Let us explain this point more clearly.
>
> Consider the example you mentioned that “If both $X_1$ and $X_2$ are direct causes of $Y$, generically they will not be independent conditional on $Y$ and $E$”. This is absolutely true. In this case, $X_1$ and $X_2$ will be coupled together and treated as one variable. Without loss of generality, let us assume that $X_2$ is absorbed into $X_1$. Similarly, if $Y$ has more than two direct causes, all the other causes will be absorbed into $X_1$. Now, the question goes to how to represent the variable $X_1$ so that it is flexible enough to contain the multiple direct causes of $Y$.
>
> If the data is simple, it is enough that $X_1$ is a one-dimensional continuous variable.
>
> If the data is complex, we can let $X_1$ be a multi-dimensional continuous variable, say m-dim. Further, for simplicity we can assume that all $X_i$ is a m-dimensional variable. In this case, our approach will not change except replacing one-dimensional $X_i$ with m-dimensional $X_i$.
>
> We have updated Section 3.2 to make it clearer.
>
>
>
> **Question 2.**
>
> "*Theorem 4 is less than fully rigorous and is misleading. For example, the proof of Theorem 4 invokes a method for inferring causal directions in Zhang et al. (2017), but as far as I know, that method does not yet have a rigorous theoretical justification.*"
>
> **Authors' Response**:
>
> Thank you for the comment. In order to avoid such a kind of confusion, we have clarified in the revision that the method for inferring causal directions in Zhang et al. (2017) is a heuristic one without a rigorous theoretical justification yet.

---

### Decision · Program_Chairs · 2021-01-07
**Final Decision**

**Decision:**

Reject

**Comment:**

The paper aims to provide a framework for learning non-linear feature mappings such that are invariant to environments. The critical concern raised by the reviewers is their assumption: that causal features of the label are conditionally independent given the label. But in any DAG, conditioning on a common child (here, the label) renders the parents dependent. Their assumption thus is not going to hold other than on a measure zero set of parameters.